# MALICE: Memory-aware Loop Invariant Generation on Symbolic Execution Traces

Tong Chen [* 1]  Siyu Liu [* 2]  Hongyi Zhong [* 1]  Liao Zhang [1]  Lixiang Wang [1]  Xiwei Wu [1]  Junchi Yan [1]  Qinxiang Cao [1]

## Abstract

Automatic loop invariant generation remains a challenging problem in program verification, particularly for memory-manipulating programs where shape invariants are required to characterize heap-allocated structures and memory layouts. While existing approaches succeed on numerical invariants, they achieve limited accuracy on shape invariants. We hypothesize that this stems from the need to reason about memory state evolution—information that remains implicit in source code. To address this, we ground LLM reasoning in symbolic execution traces that explicitly capture such transitions. We propose MALICE, a two-stage framework incorporating these traces: (1) guided multi-step reasoning that predicts invariants via chain-of-thought analysis of traces, and (2) agentic iterative refinement that corrects candidates through verification tool feedback. Evaluated on LIG-MM+, a benchmark featuring common operations on typical memory structures, MALICE substantially outperforms existing approaches.

## 1. Introduction

Program verification is the cornerstone of modern software infrastructure, playing a significant role in safety-critical domains including operating systems, compilers, and financial systems. It has also attracted increasing attention in the context of trustworthy LLM code generation. Many traditional verification approaches utilize assertions to describe critical program states, reducing program correctness to assertion entailment. However, traditional methods often struggle to scale with the sophisticated memory management patterns and complex data structures prevalent in contemporary software applications. Central to this verification paradigm are loop invariants—conditions that hold true before and after each iteration of a loop. These invariants serve as essential checkpoints, capturing fundamental properties preserved throughout execution.

Specifically, verifying memory-manipulating programs requires reasoning over two primary types of invariants:

- **Data Invariants**: Ranging from simple numerical relations to more complex constraints over different data types.
- **Shape Invariants**: Characterizing the structural properties of heap-allocated data (e.g., list segments) to ensure memory safety.

While verifying functional correctness requires the integration of both, generating valid shape invariants remains the primary bottleneck. Unlike data invariants, shape invariants must capture complex heap-allocated memory structures, which requires a profound understanding of the program's memory layout and logical dependencies. Without solving the challenge of shape analysis, ensuring functional correctness for memory-manipulating programs is impossible. Consequently, automating the generation of shape invariants has become a critical prerequisite for scalable verification.

Current research on loop invariant generation primarily falls into three categories: traditional symbolic reasoning (Padhi et al., 2016; Echenim et al., 2019), learning-based techniques (Garg et al., 2016; Si et al., 2018b), and Large Language Model (LLM) adaptations (Wen et al., 2024; Cao et al., 2025). Traditional symbolic approaches and learning-based techniques attempt to automate invariant inference with predefined patterns. LLM-based approaches treat the LLM as a proposer and employ verification tools to check the correctness of the generated invariant.

However, most of these works share a common limitation: they are predominantly designed for and evaluated on numerical benchmarks (*e.g.*, HOLA (Dillig et al., 2011), SV-COMP (Beyer, 2012)). They fail to capture the structural properties of heap-allocated data structures, such as

---

[*]Equal contribution  [1]Shanghai Jiao Tong University [2]Zhiyuan College, Shanghai Jiao Tong University. Correspondence to: Junchi Yan <yanjunchi@sjtu.edu.cn>, Qinxiang Cao <caoqinxiang@sjtu.edu.cn>.

*Proceedings of the 43$^{rd}$ International Conference on Machine Learning*, Seoul, South Korea. PMLR 306, 2026. Copyright 2026 by the author(s).

linked lists and arrays, leaving a significant gap in verifying memory-manipulating programs. While leading LLM-based frameworks (Liu et al., 2024) have begun to address memory manipulation, they still struggle to synthesize precise shape invariants involving separation logic predicates, yielding a low accuracy of 41.92%, which is insufficient for the demands of real-world system software.

We propose MALICE[1], a novel framework designed to automate the generation of shape invariants. Instead of inferring memory states solely from static source code—where memory evolution remains implicit—we ground the LLM's reasoning in symbolic execution (SE) traces. Here, symbolic execution traces are logical descriptions of program state sequences generated by symbolically executing the program. These traces naturally encode logical dependencies and memory layouts. Moreover, we incorporate natural language understanding into the analysis of SE traces. By integrating these traces into the LLM's context, we enable the model to understand how memory states evolve, significantly improving its ability to reason about structure. Lastly, to address the challenge where a single inference pass often fails to capture precise memory separation logic, we reframe the generation process from a static prediction task to a dynamic, agentic workflow. The workflow proceeds as follows:

- **Prediction**: We convert C programs into symbolic execution traces using QCP (Wu et al., 2026), a symbolic execution tool for program verification. We format the loop information with these traces to build prompts that enable the LLM to generate candidate invariants. This generation task is divided into three components: Natural Language Reduction (code understanding), Memory Assertions (separation logic for structure), and Non-Spatial Conditions (arithmetic properties).
- **Refinement**: We employ QCP to validate the synthesized invariants. If validation fails, the LLM first analyzes the error log to explain the underlying reason in natural language, and then provides a concrete revision based on the explanation and error log. The revised invariant is fed back for subsequent refinement if it remains incorrect. This cycle continues until verification succeeds or a limit is reached.

To better evaluate our results, we extend the LIG-MM benchmark (Liu et al., 2024) to LIG-MM+. Our experimental results confirm the effectiveness of our method, showing that MALICE improves the generation of precise invariants for complex memory-manipulating programs.

---

[1]Code available at https://anonymous.4open.science/r/905DAKWLC

## 2. Preliminaries

### 2.1. Hoare Logic and Symbolic Execution

Hoare Logic (Hoare, 1969) is an axiomatic system designed to prove the partial correctness of programs. The basic formulas of this logic are Hoare triples, denoted as $\{P\}C\{Q\}$. This signifies that if the execution of code $C$ begins in a state satisfying the precondition $P$ and subsequently terminates, any state reached upon completion will satisfy the postcondition $Q$.

To automate the verification of such triples via forward analysis, the Strongest Postcondition (SP) is utilized. The term $sp(P, C)$ characterizes the set of all possible states reachable by the program $C$ starting from any initial state satisfying $P$, adhering to the following property:

$$\models \{P\}C\{Q\} \iff sp(P,C) \implies Q$$

For loop-free programs, the computation of $sp$ is defined by structural recursion. For instance, the rule for sequential composition is $sp(P, S_1; S_2) = sp(sp(P, S_1), S_2)$.

In practice, Symbolic Execution implements this logic along execution paths. Conceptually, it systematically explores possible execution paths by executing the program on symbolic inputs rather than concrete ones. For a specific path $\tau$, the execution engine dynamically accumulates constraints to compute a symbolic state that encodes the strongest postcondition $sp(P, \tau)$. Accordingly, the validation of the triple $\{P\}\tau\{Q\}$ reduces to checking the entailment $sp(P, \tau) \implies Q$, which is termed Verification Condition (VC). A constraint solver is then employed to check the satisfiability of its negation.

Figure 1 illustrates this process. Starting from an initial state $S_0$, the symbolic state evolves via strongest postconditions (e.g., $S_1 := sp(S_0, C_1)$). At branching points, the execution splits, and each path independently accumulates the corresponding branch condition. Although the control flow merges at the end, engines typically analyze each path independently.

We formally define the linear history of state evolution along a single path as a *symbolic execution trace*:

$$\{S_0\}C_1\{S_1\}C_2\{S_2\}\ldots C_n\{S_n\}$$

Here, each $C_i$ is an execution unit. In different scenarios of program verification and program analysis, these $C_i$ can represent program units at varying levels of granularity. For example, for fine-grained program analysis, each $C_i$ would be a single assignment command, and such a symbolic execution trace describes the program states after executing every single command. In loop analysis, $C_i$ can represent an entire loop body, which allows traces to capture state changes across each complete loop as well.

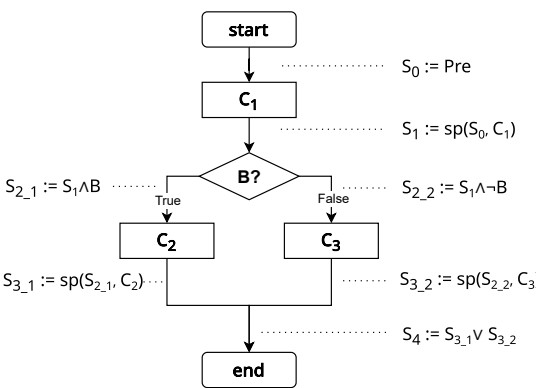

*Figure 1.* An illustrative example of symbolic execution on a Control Flow Graph.

Formally, program correctness is established by verifying that the final state $S_n$ of each such trace entails the desired postcondition $Q$.

However, applying symbolic execution to unbounded loops is often infeasible due to path explosion. This necessitates the use of loop invariants—assertions that characterize the loop's behavior and remain valid before and after each iteration. Formally, for a program segment comprising a loop `while B do S` followed by a program suffix $S_1$, the inference rule is given by:

$$\frac{(1)\ P \implies Inv \quad (2)\ \{Inv \wedge B\}\ S\ \{Inv\} \quad (3)\ \{Inv \wedge \neg B\}\ S_1\ \{Q\}}{\{P\}\ \textbf{while } B\ \textbf{do}\ S; S_1\ \{Q\}}$$

Consequently, establishing both the correctness of the triple and the validity of the invariant reduces to checking these three conditions: (1) Initialization: $P \implies Inv$ (2) Preservation: $\{Inv \wedge B\}S\{Inv\}$ (3) Exit: $\{Inv \wedge \neg B\}S_1\{Q\}$

### 2.2. Verification based on Separation Logic

Separation Logic (Reynolds, 2009) extends classical Hoare logic to reason about programs with mutable data structures. Central to this extension is the *separating conjunction* operator $*$, which enforces that no memory location is shared between subcomponents of the heap. The assertion $P * Q$ denotes that the memory state can be partitioned into two disjoint parts, satisfying $P$ and $Q$ respectively. Crucially, this enforces strict resource ownership: unlike classical logic where $P \wedge P \iff P$, the formula $P * P$ is unsatisfiable for non-empty heaps, as it implies conflicting ownership of the same memory address.

Usually, separation logic assertions are represented in the symbolic heap normal form:

$$\exists \overrightarrow{x}.\ (P_1 \wedge P_2 \wedge \cdots \wedge P_n) \wedge (Q_1 * Q_2 * \cdots * Q_m),$$

where the *pure* component (the $P_i$) captures logical constraints over variables independent of the heap, and the *spatial* component (the $Q_j$) consists of a separating conjunction of heap-related predicates.

To operationalize the generation and verification of such memory-safety assertions, we adopt the *Qualified C Programming (QCP)* framework (Wu et al., 2026). We selected QCP specifically because its verification engine is natively grounded in the separation logic framework described above. It performs symbolic execution to verify C programs, generating traces of symbolic state evolution and other associated outputs. In these traces, heap structures are represented using spatial predicates like `data_at` for memory cells and `lseg` or `listrep` for linked lists.

## 3. Methodology

This section presents our methodology for automatically generating memory-aware loop invariants. The main idea is to leverage both the LLM's comprehension of C program behavior and the symbolic tool QCP's capability to provide precise feedback. Subsequently, a refinement mechanism is introduced to correct subtle errors that the LLM might produce.

The discussion initially centers on the framework in the context of single-loop scenarios, elucidating the design details, and subsequently extends this approach to handle functions containing multiple loops that therefore require multiple invariants.

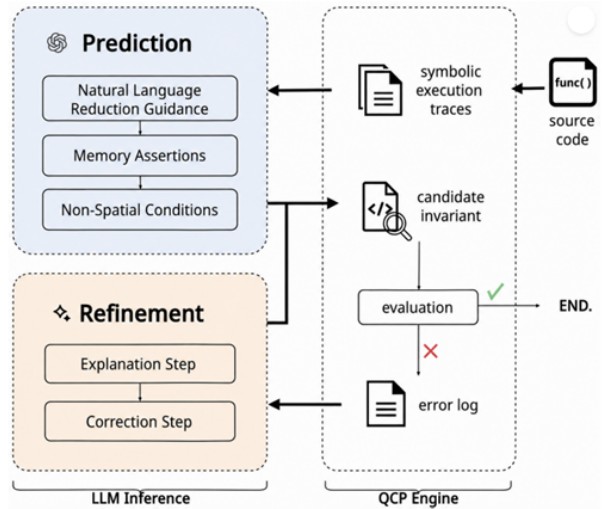

*Figure 2.* Workflow of the single-loop invariant generation framework.

## 3.1. Core Framework Overview

Figure 2 shows the core workflow of loop invariant generation for single-loop programs. It comprises two main components:

**Prediction.** This process aims to synthesize candidate loop invariants by formulating a task-specific context. Using the prediction context, which comprises the predicate selection plan, memory assertions, and pure properties, the system executes a chain of three LLM interactions. The model first analyzes code semantics to infer memory permissions and select representative predicates. Second, leveraging these chosen predicates and symbolic traces, the model performs assertion analysis and reduction to synthesize memory assertions. Finally, it derives non-spatial constraints (e.g., arithmetic constraints), which are conjoined with the memory assertions to constitute the final candidate invariant.

**Refinement.** This stage aims to refine candidate loop invariants that may miss one or two critical terms. The process is implemented in two steps. The LLM is first asked to briefly interpret the lengthy execution feedback in natural language to diagnose the root cause. Subsequently, it utilizes both the interpretation and execution feedback to revise the invariant.

Once the Prediction phase synthesizes a candidate invariant, it is injected into the program and subjected to formal verification by QCP. If this verification passes, the process completes successfully. Conversely, if verification fails, the system initiates the refinement phase, where it iteratively refines the previous incorrect candidate until a valid invariant is found or the maximum retry limit is reached.

In designing this workflow, we observe that naive prompting strategies frequently struggle to capture complex topologies directly from source code or ensure the spatial disjointness required by separation logic, often leading to redundant or omitted memory regions. To mitigate these structural errors during the Prediction and Refinement phases, we construct two artifacts derived from symbolic execution: symbolic execution traces and loop unrolling information.

**Fine-grained Symbolic Information.** This information denotes the fine-grained traces described in Section 2.1, which capture the step-by-step state evolution between simple statements within the QCP engine. In loop analysis, it typically refers to state transitions within the loop body. During the Refinement phase, these traces enable the model to traverse the execution history and pinpoint the exact state immediately preceding a verification failure. By investigating why the symbolic state failed to satisfy the expected safety properties, the agent can effectively interpret the root cause of the error and deduce the necessary amendments to the invariant.

**Loop Unrolling Information.** To explicitly capture the

state evolution across loop iterations, we unroll the loop and utilize the resulting loop execution traces to inform the prediction phase. Formally, for a loop `while(B)C`, we define the single iteration step as $T \triangleq \mathtt{assume}(B); C$. By unrolling this step for $K$ iterations, we generate the sequential trace $\{S_0\}\ T\ \{S_1\}\ \ldots\ T\ \{S_K\}$. Here, $S_0$ denotes the loop entry state, and $S_i$ represents the symbolic state after the $i$-th execution of $T$. We posit that analyzing these explicit state transitions enables LLMs to discern the loop's regularity, allowing them to infer the inductive logic that characterizes its infinite behavior. In practice, the engine generates this trace by modeling the unrolled loop as a sequence of `if(B)C` blocks and yielding the states $S_0, \ldots, S_K$, as shown in Figure 3.

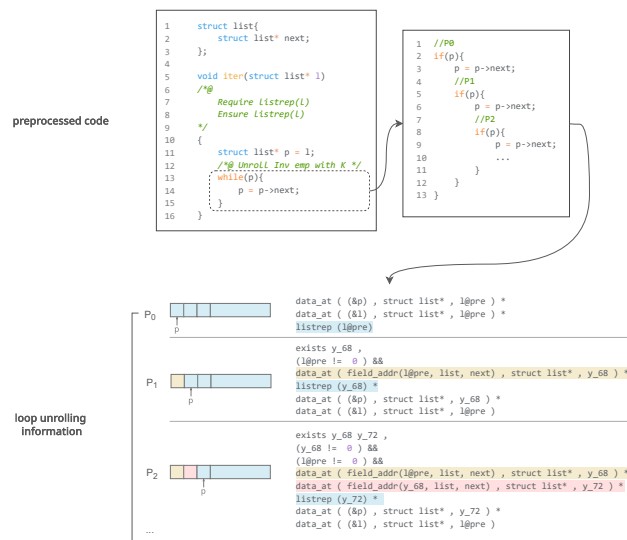

*Figure 3.* Unrolling information of singly-linked list traversal.

## 3.2. Trace-Based Generation

This subsection outlines the three stages of the generation process, all of which are implemented via In-Context Learning (ICL).

### 3.2.1. ABSTRACT REDUCTION GUIDANCE

Although direct invariant generation may yield inaccuracies, LLMs' extensive pre-training on C corpora grants them a robust understanding of code semantics and logic. Given this capability, we hypothesize that first having the LLM generate an abstract reduction guidance to steer the analysis of the reduction materials is more tractable than directly generating invariants from these materials alone.

To operationalize this, we design a Chain-of-Thought (CoT) prompt comprising four stages:

1. summarize the loop's high-level functionality and explicitly reason about read/write permissions.

2. exhaustively enumerate all active variables and heap structures to construct a complete snapshot of the current memory state.

3. dynamically partition the accessed memory state into distinct logical regions to capture the structural evolution inherent in loop traversals (e.g., splitting a list into "traversed" and "untraversed" segments).

4. map the identified regions to specific abstract predicates and explicitly define boundaries.

This reasoning chain ultimately forms a structured intermediate plan. For instance, the predicate selection plan for the traversed region is illustrated below:

---

**Snippet: Predicate Selection Plan**

**Traversed Region** ($l \rightarrow p$)**:**
- *Shape:* Sequence of nodes traversed from head $l$.
- *Boundary:* Excludes current cursor $p$.
- *Predicate:* Maps to `lseg`.

---

*Figure 4.* Snippet of the structured intermediate plan for singly-linked list traversal function: mapping a memory region to a formal predicate.

### 3.2.2. TWO STRATEGIES FOR MEMORY ANALYSIS

Based on reduction guidance and loop unrolling, we propose two strategies to guide LLMs in generating memory predicates:

The first strategy, **Reduction on Single Assertion**, is motivated by the insight that assertions derived from deeper loop unrolling information often exhibit more discernible recursive patterns. In particular, for memory assertions, the incremental differences between iterations are explicitly manifested in the trace.

Considering $P_3$ in Fig. 3, the two nodes already traversed by pointer `p` are clearly manifested as two consecutive `data_at` predicates. Guided by the predicate plan in Figure 4, we reduce these predicates to `lseg(l@pre, y_72)`, denoting the list segment from `l` to `p`. By folding such concrete heap segments into recursive predicates within the assertion, we effectively abstract away the specific unrolling depth, thereby directly yielding a general loop invariant.

Additionally, limiting analysis to a single assertion reserves sufficient prompt capacity for the LLM to scrutinize memory boundaries, thereby facilitating precise parameter inference and enabling predicate correction.

The second strategy is **Generalization on State Traces**, operating on the principle that every reachable state implies the loop invariant. In contrast to the first method, which prunes a state to adapt to a target, this approach aligns multiple state instances to discover their shared structure.

Guided by a mapping from memory to predicates, we verify whether a spatial shape persists across all aligned instances simultaneously. Continuing to use Fig. 3 and Fig. 4 as our example, the guidance in Fig. 4 directs the abstraction of the region from `l` to `p` into a list segment. Corresponding concrete segments are identified in Fig. 3 across $P_0$ to $P_2$– manifesting as an empty heap, a single-node permission, and two-node permissions, respectively. Since the spatial shape holds despite these size differences, we successfully reduce the region to the generalization `lseg(l, p)`.

A test case passes if either of the two strategies works for that case.

### 3.2.3. INFERENCE OF PURE PROPERTIES

We employ a hypothesize-and-verify strategy to derive pure properties. The process starts by proposing candidate logical relations, such as numerical bounds, pointer validity, or variable relationships. These candidates are then subjected to trace-based verification, where they are checked against concrete traces. Only properties that consistently hold across all observed traces are retained to complement the spatial invariant.

### 3.3. Two-step Refinement

The refinement phase optimizes loop invariants by leveraging runtime feedback from the symbolic execution engine. The feedback contains the following ingredients:

- basic information such as the error type and error location
- the program state immediately preceding the failure, which indicates the most direct cause of failure.
- the entire symbolic execution trace
- possibly disproved verification gaps. For example, in the case of "Entailment Check Failed", it reports the unproven entailment together with a natural-language explanation.

To begin with, the framework is implemented through few-shot prompting, where we manually design a prompt template. In the Explanation step, it mainly follows the steps of identifying the error type, locating the error position, and explaining why the failed entailment cannot be automatically proven by analyzing the detailed memory topology. In the Correction step, the LLM is prompted to restate and thereby comprehend the explanation from the previous step and produce a concrete revised loop invariant. As few-shot examples, we case study several incorrect loop invariants for a function that appends two doubly-linked lists and a function that copies an array to another address, and we manually write case analyses for them following the prompt template above.

These cases cover several common error types, mainly consisting of **Syntax Errors**, **Entailment Check Failures**, and **Numerical Overflow Errors**.

Moreover, we discovered that the Refinement task is highly amenable to synthesizing both unsupervised and supervised training data, since generating incorrect candidates is essentially free and their advisory revisions can be distilled via the aforementioned ICL process. This creates opportunities for Supervised Fine-Tuning (SFT) and reinforcement learning algorithms such as DPO or GRPO on smaller models. We choose SFT in this work, but similar idea extends to DPO. To rival the ICL capabilities of teacher models, we implement a data filtration strategy: if an iterative refinement process passes at iteration $K$, then we regard the intermediate corrections made at iterations 1 through $K - 1$ as preferred data and discard all others. Experimental results are discussed in the Experiments section.

### 3.4. Hierarchical Organization of Loop Invariants

When a function contains multiple loops, generating their invariants demands a structured approach. Since obtaining unrolling information for a specific loop depends on valid symbolic execution of the preceding code, loops must be organized according to symbolic execution flows. Specifically, cascaded loops are processed in linear order, while nested structures require the inner loop to be resolved prior to the outer loop. To enforce this order, we index loops by their occurrence, organize them into a hierarchical tree as detailed in Algorithm 1, and generate the final execution sequence via a post-order traversal.

---

**Algorithm 1** Loop Dependency Tree Construction

1: **Input:** Ordered events $\mathcal{E}$ from code (Loops and Scope Delimiters)
2: **Output:** Dependency Tree Root $\mathcal{T}_{root}$
3: Initialize $\mathcal{T}_{root}$ and active node map $\mathcal{M}$ with $\mathcal{M}[-1] \leftarrow \mathcal{T}_{root}$
4: Initialize current depth $d \leftarrow 0$
5: **for** each event $e \in \mathcal{E}$ **do**
6:    **if** $e$ is a Scope Delimiter ($\{, \}$) **then**
7:       Update depth $d$ (increase on $\{$, decrease on $\}$)
8:    **else if** $e$ is a Loop Header **then**
9:       Create node $n$
10:      $n.parent \leftarrow \mathcal{M}[d - 1]$
11:      $n.parent.\text{addChild}(n)$
12:      $\mathcal{M}[d] \leftarrow n$
13:    **end if**
14: **end for**
15: **return** $\mathcal{T}_{root}$

---

## 4. Experiments

In this section, we evaluate our proposed framework on LIG-MM+, a benchmark we build by extending the previous LIG-MM benchmark (Liu et al., 2024) to evaluate loop invariant generation for memory-aware real-world C programs.

### 4.1. LIG-MM+ Benchmark

The previous benchmark **LIG-MM** (Liu et al., 2024) collects C functions on various data structures from several real-world sources. However, we noticed that **LIG-MM** does not cover typical array operations, which are ubiquitous in real-world system software and require reasoning over contiguous memory addresses, making them fundamentally different from pointer-based structures. Therefore, we propose LIG-MM+, which adds array manipulation programs (e.g., `array_max`, `array_sum`, `array_copy`) to LIG-MM.

Table 1 summarizes the statistics of LIG-MM+ by category, including the number of programs, source repositories or benchmarks, and covered data structures.

*Table 1.* Statistics of the LIG-MM+ benchmark

| Category | # Programs | Average Code Lines | Data Structures |
|----------|-----------|--------------------|-----------------|
| DLL | 133 | 36 | Doubly linked list |
| SLL | 96 | 30 | Singly linked list |
| Array | 8 | 21 | Array |
| Others | 23 | 35 | Tree, hash table |
| Overall | 260 | 35 | (omitted) |

We emphasize that all programs in our dataset involve at least one data structure and at least one loop, and sometimes two loops. None of the programs is purely numerical, which means the specification of each program requires memory safety and correctness.

### 4.2. Experimental Setup

We compare our framework against the following baselines:

- **AutoSpec** (Wen et al., 2024): An LLM-based specification generation tool.
- **SLING** (Le et al., 2019): A traditional SMT-based approach to inferring loop invariants in separation logic.
- **LIG-SE**: A loop invariant generation method based on LLM and symbolic execution.

We choose DeepSeek-V3.2 as our default backbone LLM, except when comparing backbone models in the ablation study. For the two memory analysis strategies introduced in Section 3.2.2, the final set of passed test cases is the union of the passed sets produced by both strategies. We set the loop unrolling depth to 4 and the maximum number of refinement iterations to 10.

### 4.3. Main Results

Table 2 presents the comparison of our approach with the baselines.

*Table 2.* Comparison of Loop Invariant Generation Performance (Pass Rate@8)

| Method | Overall Pass Rate |
|---|---|
| AutoSpec | 0.00% |
| SLING | 26.15% |
| LIG-SE | 41.92% |
| MALICE (ours, ICL) | 82.69% |
| MALICE (ours, SFT) | 86.15% |

MALICE(ICL) refers to implementing the Refinement stage via in-context learning (ICL), while MALICE(SFT) refers to implementing the Refinement stage via supervised fine-tuning (SFT). A more detailed discussion of SFT is provided in Section 4.5.

The experimental results show that our method surpasses all baselines.

We remark that AutoSpec cannot handle memory-aware program verification because it was designed for numerical programs. LIG-SE does not support non-inductive data structures such as arrays because its self-supervised learning stage requires inductive expansion of predicates for the data structure. SLING does not support continuous address offsets when designing predicates for array structures. In contrast, MALICE reaches 100% on the Array category.

Specifically, for several multi-loop programs in our dataset, our method reaches 100% accuracy in the pass@8 test.

### 4.4. Ablation Study

We conducted an ablation study to evaluate the impact of different components of our prompt and pipeline. The variants considered are:

- **Vanilla**: We minimally prompt the model, asking for loop invariants given the code and a tutorial on the QCP tool as domain knowledge.
- **Backbone LLM**: We compare the performance of our framework with various backbone LLMs, including **DeepSeek-V3.2**, **Qwen3-Max**, and **DeepSeek-V3.2-Thinking**, covering both instruction and reasoning models.
- **w/o Prediction**: We replace the entire Prediction stage in our framework with the same minimal prompt used in the **Vanilla** setting, followed by the Refinement stage.
- **w/o Refinement**: We disable the iterative refinement loop and rely solely on the first-pass generation.
- **Binary Feedback Only**: We examine how important

the symbolic trace and QCP feedback are during the Refinement stage compared with the iterative framework itself. In this experiment, each refinement iteration is provided with only the pass/fail status, without the error position or type from QCP, the symbolic execution trace, or detailed error analysis. The iteration depth ($K = 10$) and pass@8 evaluation are kept the same as in the full pipeline.

- **Zero Shot**: We remove the few-shot examples provided in the prompt.

#### 4.4.1. BACKBONE LLMs

The performance comparison of our framework with various backbone LLMs is summarized in Table 3.

*Table 3.* Performance Comparison of Various Backbone LLMs (Pass Rate@8)

| Model | Overall Pass Rate |
|---|---|
| DeepSeek-V3.2 | 82.69% |
| Qwen3-Max | 75.00% |
| DeepSeek-V3.2-Thinking | 82.69% |

From the results, our method demonstrates general applicability across various state-of-the-art backbone LLMs, from instruction models to reasoning models. We choose DeepSeek-V3.2 as our backbone LLM because it achieves top performance while maintaining token efficiency.

#### 4.4.2. ABLATION STUDY OF INVOLVED STAGES

Table 4 summarizes the rest of the ablation study.

*Table 4.* Ablation Study Results (Pass Rate@8)

| Method | Overall Pass Rate |
|---|---|
| Full Pipeline (DeepSeek-V3.2) | 82.31% |
| Vanilla | 15.77% |
| w/o Prediction | 66.15% |
| w/o Refinement | 34.62% |
| w/o Refinement (even pass@150) | 47.31% |
| Binary Feedback Only | 36.15% |
| Zero Shot | 41.15% |

According to the experimental results, the pass rate of **Full Pipeline** considerably surpasses **Vanilla**, indicating that our framework indeed improves the LLM's performance on program verification tasks through nontrivial prompt engineering and workflow design. Moreover, comparing **Full Pipeline**, **w/o Prediction**, and **w/o Refinement (even pass@150)**, we conclude that the Refinement stage is more important than the Prediction stage, and that Refinement does not simply work since it outputs 10 times more tokens. Furthermore, comparing **Binary Feedback Only** and **w/o Refinement**, keeping only the iterative framework with pass/fail binary feedback during the Refinement stage

performs only marginally better than having no Refinement stage; both dramatically underperform the full pipeline. This shows that symbolic traces and feedback provide substantial benefits. Finally, comparing **Full Pipeline** and **Zero Shot**, we conclude that few-shot prompting is vital in the Prediction and Refinement stages.

### 4.5. Refinement with SFT

In this section, we implement the Refinement stage using SFT instead of ICL. The SFT corpus is distilled from ICL refinement trajectories generated by MALICE, where DeepSeek-V3.2 serves as the teacher model. We retain only trajectories that eventually lead to successful invariants to maintain high-quality data. For each program that succeeds at refinement iteration $K$, we extract iterations $1$–$(K-1)$ as training examples. This process yields 9,698 training samples in total. The input-output pair of each sample is the same as in the ICL refinement stage: the source code, the previous incorrect invariant, and the QCP feedback serve as input, and the target output is a natural-language explanation that diagnoses the root cause of the failure, followed by a concrete correction containing the revised invariant.

We use Meta-Llama-3.1-8B-Instruct as the base model and fine-tune it with LoRA implemented in LLaMA Factory. More training configuration details are provided in the appendix. To avoid high variance in the intrinsic verification difficulty of different programs and prevent data leakage, we adopt $k$-fold cross-validation with $k$ set to 5. Specifically, we group all programs into five folds and gather all refinement trajectories of each program into its corresponding fold. We then evaluate each fold on the model trained with data from the other four folds. Training takes about 18 hours on one NVIDIA RTX 5090 GPU per fold.

The experimental results are summarized in Table 5.

*Table 5.* Performance of SFT (Pass Rate@8)

| Method | Overall Pass Rate |
|--------|-------------------|
| SFT | 86.15% |
| ICL | 82.69% |

In the experimental results, the performance of SFT on the 8B Llama-3.1 model surpasses that of ICL on the 685B DeepSeek-V3.2 model. This shows the great potential of training models on the Refinement task to enhance program verification performance.

### 4.6. Fairness Concern

**Model Scale**  We further analyze whether MALICE's advantage mainly comes from using a larger backbone model than other LLM-based baseline methods such as LIG-SE. As shown in Table 4, under the same 685B DeepSeek-V3.2

backbone, the Vanilla setting achieves only 15.77%, which is still far below LIG-SE's 41.92% with CodeGen-350M. In contrast, MALICE (ICL) reaches 82.69%, giving a 66.92 percentage-point improvement over Vanilla. This comparison isolates the effect of our prediction-and-refinement framework under the same model scale.

We further conduct a new Full-8B experiment to remove the dependency on large models entirely. In this setting, the Prediction stage uses Qwen3-8B with in-context learning, while the Refinement stage uses the fine-tuned Llama-3.1-8B model presented in Section 4.5. As shown in Table 6, Full-8B achieves 81.54%, substantially outperforming LIG-SE and approaching the performance of MALICE with DeepSeek-V3.2. Moreover, MALICE (SFT) already uses an 8B model for Refinement and achieves 86.15%, surpassing the 685B ICL variant. These results indicate that MALICE's effectiveness is not merely a consequence of using a larger backbone model.

*Table 6.* Model-scale analysis (Pass Rate@8)

| Method | Prediction | Refinement | Pass Rate |
|--------|-----------|-----------|-----------|
| LIG-SE | CodeGen-350M (SFT) | – | 41.92% |
| Full-8B (New) | Qwen3-8B (ICL) | Llama-3.1-8B (SFT) | 81.54% |
| MALICE (SFT) | DeepSeek-V3.2 | Llama-3.1-8B (SFT) | 86.15% |
| MALICE (ICL) | DeepSeek-V3.2 | DeepSeek-V3.2 | 82.69% |

**Model Query Times**  The full MALICE pipeline calls the LLM 3 times in the Prediction stage and up to 10 times in the Refinement stage during each run. With pass@8 evaluation, it queries the LLM up to $8 \times (3 + 10) = 104$ times. We ask whether the MALICE workflow can surpass the baseline under the same LLM query budget. Toward this, we further conduct a pass@1 experiment where we decrease MALICE's maximum refinement depth to 5, thus querying the LLM 8 times overall. Table 7 compares this experiment with the baseline pass@8 LIG-SE setting under the same 8-call budget.

*Table 7.* Comparison under explicit model-query budgets

| Method | Pass@ Times | Per-Run Behavior | Pass Rate |
|--------|-------------|------------------|-----------|
| LIG-SE | 8 | LLM + SE feedback loop | 41.92% |
| MALICE w/o Refinement (even pass@150) | 150 | 3 LLM calls (prediction only) | 47.31% |
| MALICE (1×8 calls, New) | 1 | 3 pred. + 5 refine. = 8 total calls | 59.62% |
| MALICE Full | 8 | 3 pred. + up to 10 refine. | 82.69% |

The single-run MALICE setting with the same total number of eight LLM calls already surpasses pass@8 LIG-SE, showing that MALICE's gain does not come merely from repeating independent samples. Conversely, MALICE w/o Refinement with pass@150 reaches only 47.31%, barely exceeding LIG-SE and underperforming $1 \times 8$ calls MALICE. These results indicate that the workflow, especially the refinement stage, rather than brute-force repetition, is essential to the final performance.

## 5. Related Work

**Traditional Approaches.** Traditional methods for loop invariant generation often rely on symbolic analysis and program synthesis. Tools like LoopInvGen (Padhi et al., 2016) and Llinva (Echenim et al., 2019) use precondition inference and SMT-based condition strengthening, respectively. (Yang & Sergey, 2025) extends the inductive logic programming tool Popper (Cropper & Morel, 2021) to utilize negative examples for invariant search. While effective for numerical invariants, these approaches generally struggle to handle the structural complexities of heap memory. Although SLING (Le et al., 2019) attempts to address this by applying dynamic analysis to infer inductive shape properties, it incorporates hard-coded algorithms tailored to specific data structures and is therefore not flexible enough for more general problems.

**Learning-based Approaches.** Learning-based approaches have drawn more attention recently due to their flexibility and substantial performance. ICE-DT (Garg et al., 2016), CODE2INV (Si et al., 2018a), and its extensions (Yao et al., 2020; Ryan et al., 2020) utilize decision trees, reinforcement learning, and neural networks to infer invariants, but they generally target numeric programs with scalar variables and often struggle to capture the structural memory semantics required for heap-manipulating programs.

**LLM Adaptations.** With the rise of Large Language Models (LLMs), researchers have begun exploring their potential for invariant generation. AutoSpec (Wen et al., 2024) introduces a framework that synergizes LLMs with static analysis and program verification, decomposing the program to guide the synthesis of specifications. Clause2Inv (Cao et al., 2025) explores the usage of LLMs' in-context learning capabilities to synthesize loop invariants without the need for fine-tuning. Despite these advancements, these methods are often constrained by the models' shallow grasp of the underlying reasoning principles associated with complex memory separation logic. Taking a different approach, LIG-MM (Liu et al., 2024) proposes a self-supervised fine-tuning paradigm that contributes a valuable dataset and establishes a baseline. However, this approach potentially underutilizes the LLMs' inherent pre-trained capabilities in comprehending C program behavior.

## 6. Conclusion and Limitations

**Conclusion** This paper introduces MALICE, a framework that incorporates symbolic execution traces to automate the generation of shape loop invariants. We designed distributed task prompts that guide LLMs to analyze these traces using natural language, and integrated an iterative

refinement mechanism to continuously correct candidates. Furthermore, we derived the LIG-MM+ benchmark from existing datasets, summarizing typical data structure behaviors to enable more comprehensive evaluation. Our evaluation on this benchmark validates the robust capability of MALICE in synthesizing complex shape invariants.

**Limitations** Despite these encouraging results, MALICE still has several limitations. First, generalizing shape properties from symbolic execution traces remains challenging: traces expose heap configurations only after a finite number of loop unrollings, and the LLM must infer an inductive abstraction from them. This difficulty is amplified when programs manipulate complicated data structures or combine several data-structure idioms. Moreover, to better generalize the method to a broader range of real-world programs and evaluate its effectiveness more comprehensively, the benchmark needs to include richer data structures and more diverse loop patterns. Although LIG-MM+ extends the previous LIG-MM benchmark in this respect, its current scale and diversity are still insufficient to fully reflect the complexity of real-world verification tasks due to huge human labor cost of labelling each program's formal specification. Finally, MALICE is set to call up to 10 sequential iterative refinement steps in each run. The time and resource consumption is fairly large. We therefore seek mechanisms to parallelize the procedure or decrease LLM calls while preserving performance.

## Impact Statement

This paper presents work whose goal is to combine formal methods with machine learning to provide trustworthy code generation. There are many potential societal consequences of our work, none of which we feel must be specifically highlighted here.

## Acknowledgements

This work was supported by the NSF China (No. 62472274).

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

# A. Prompt Construction

## A.1. Prompt for Prediction Stage

Due to the complexity and length of the prompts required for the Prediction stage, we present it in a structured format rather than as raw text. Specifically, we decompose the prompt into the following reusable functional components:

**Introductory Documentation** This component provides a formal introduction to the QCP framework. It details the program state representation based on separation logic and defines the rigorous syntax specifications required by QCP.

**Task Instructions** This part outlines the fundamental task objectives, enforces a structured CoT workflow, and prescribes a strict output template. Additionally, it provides detailed analytical principles to guide specific steps of the reasoning process.

**Predicate Definitions** This segment formally defines the QCP-specific separation logic predicates. It covers primitive memory assertions (e.g., data_at) and specifications for recursive data structures, including both complete lists and segments for singly and doubly linked lists. Each entry details the predicate's semantics, parameter roles, and the precise memory layout required for verification.

**Symbolic Execution Materials** For a detailed introduction, please refer to Section 3.2.

**Illustrative Examples** This component presents an illustrative example corresponding to the specific task, serving as a few-shot demonstration to guide the model's output generation.

**Target Source Code** This part presents the specific C source code function to be verified, serving as the primary input for the reasoning process.

Table 8 summarizes the prompt composition utilized across the three sub-phases of the generation process.

*Table 8.* Composition of prompt components utilized across the generation phase ($\sqrt{}$ denotes inclusion).

| PHASE | INTRO | TASK | PRED | SYMB | EXMPL | CODE |
|---|---|---|---|---|---|---|
| ABSTRACT RED. | $\sqrt{}$ | $\sqrt{}$ | $\sqrt{}$ | | $\sqrt{}$ | $\sqrt{}$ |
| MEM. ASSERT. | $\sqrt{}$ | $\sqrt{}$ | $\sqrt{}$ | $\sqrt{}$ | $\sqrt{}$ | $\sqrt{}$ |
| PURE PROP. | $\sqrt{}$ | $\sqrt{}$ | $\sqrt{}$ | $\sqrt{}$ | $\sqrt{}$ | $\sqrt{}$ |

# B. End-to-End Example for Appendix

This section presents an end-to-end example of MALICE on the benchmark array_max. The input program contains a missing loop invariant before the for loop.

## B.1. Input Program

```c
#include "verification_stdlib.h"
#include "verification_list.h"
#include "int_array_def.h"
#include "sll_shape_def.h"

int array_max(int *a, int n)
/*@ Require 0 <= n && n < INT_MAX && IntArray::full_shape(a, n)
    Ensure IntArray::full_shape(a, n)
*/
{
  int i;
  int max = -1;
  /*@ Inv
      TO INFILL LOOP INVARIANT HERE
  */
  for (i = 0; i < n; ++i) {
    if (a[i] > max) {
      max = a[i];
    }
  }
  return max;
}
```

## B.2. Loop Unrolling Information

Given the input program, the symbolic execution tool produces loop-head assertions for bounded loop unrollings. At each unrolling depth, symbolic execution may produce multiple branch-specific states because the loop body contains a conditional statement. For readability, we show only one representative state at each depth below. MALICE uses the full set of generated states when abstracting memory assertions and validating pure constraints.

```
P0:
(0 <= n@pre) &&
(n@pre < INT_MAX) &&
data_at((&max), int, -1) *
data_at((&i), int, 0) *
data_at((&n), int, n@pre) *
data_at((&a), int*, a@pre) *
IntArray.full_shape(a@pre, n@pre)

P1:
∃ x_111,
(x_111 <= -1) &&
(0 < n@pre) &&
(0 <= n@pre) &&
(n@pre < INT_MAX) &&
data_at((&max), int, -1) *
data_at((&i), int, 0 + 1) *
data_at((&n), int, n@pre) *
data_at((&a), int*, a@pre) *
IntArray.full_shape(a@pre, n@pre)

P2:
∃ x_114 x_120,
(x_120 <= x_114) &&
((0 + 1) < n@pre) &&
(0 < n@pre) &&
(0 <= n@pre) &&
(n@pre < INT_MAX) &&
data_at((&max), int, x_114) *
data_at((&i), int, (0 + 1) + 1) *
data_at((&n), int, n@pre) *
data_at((&a), int*, a@pre) *
IntArray.full_shape(a@pre, n@pre)

P3:
∃ x_126 x_138,
(x_138 <= x_126) &&
(((0 + 1) + 1) < n@pre) &&
((0 + 1) < n@pre) &&
(0 < n@pre) &&
(0 <= n@pre) &&
(n@pre < INT_MAX) &&
data_at((&max), int, x_126) *
data_at((&i), int, ((0 + 1) + 1) + 1) *
data_at((&n), int, n@pre) *
data_at((&a), int*, a@pre) *
IntArray.full_shape(a@pre, n@pre)

P4:
∃ x_144 x_168 x_192,
(x_168 > x_144) &&
((((0 + 1) + 1) + 1) < n@pre) &&
(((0 + 1) + 1) < n@pre) &&
((0 + 1) < n@pre) &&
(0 < n@pre) &&
(0 <= n@pre) &&
(n@pre < INT_MAX) &&
data_at((&max), int, x_192) *
data_at((&i), int, (((0 + 1) + 1) + 1) + 1) *
data_at((&n), int, n@pre) *
data_at((&a), int*, a@pre) *
IntArray.full_shape(a@pre, n@pre)
```

## B.3. Prediction

MALICE decomposes the invariant prediction task into three subgoals: predicate selection, memory assertion abstraction, and pure constraint generation.

### B.3.1. STEP 1: ABSTRACT REDUCTION (PREDICATE SELECTION)

**Input.** This step takes the source program, the loop-invariant hole, and the available predicate definitions. The relevant source fragment is the `for` loop in `array_max`, and the relevant precondition is `IntArray::full_shape(a, n)`.

**Analysis Chain.** MALICE first identifies the target loop:

```
for (i = 0; i < n; ++i) {
  if (a[i] > max) {
    max = a[i];
  }
}
```

It then analyzes the memory accesses in the loop body. The test `a[i] > max` reads the array element `a[i]` and the scalar variable `max`. If the branch is taken, the assignment `max = a[i]` reads the same array element again and writes the scalar variable `max`. The loop update `++i` writes the scalar variable `i`. Therefore, a loop-head invariant must preserve permissions for the scalar variables `i` and `max`, and it must also preserve enough array permission to justify the access to `a[i]`.

MALICE then compares the loop structure with the precondition. The precondition gives a single full-array predicate:

```
IntArray::full_shape(a, n)
```

This predicate is sufficient at function entry, but it does not expose the moving boundary created by the loop index. At a loop head, `i` separates the array into two regions:

- a processed prefix containing indices `0` through `i - 1`;
- an unprocessed suffix containing indices `i` through `n - 1`.

This boundary is important because the current array access `a[i]` belongs to the unprocessed suffix, while previously visited elements belong to the processed prefix.

**Output.** The generated output of Step 1 is a mapping from memory components to abstraction predicates:

- Use `data_at` to abstract variables `i` and `max`. This part contains the permissions of variables `i` and `max`, and does not contain array permission.
- Use `IntArray::ceil_shape` to abstract the processed array segment. This part contains the permissions of array elements with indices from `0` to `i - 1`.
- Use `IntArray::ceil_shape` to abstract the unprocessed array segment. This part contains the permissions of array elements with indices from `i` to `n - 1`.

### B.3.2. STEP 2: MEMORY ASSERTION ABSTRACTION

**Input.** This step takes three inputs: the predicate-selection result from Step 1 ,the loop-unrolling information and predicate usage document. The predicate-selection result specifies three reduction targets: scalar variables, the processed array segment, and the unprocessed array segment.

**Analysis Chain.** MALICE inspects the loop-head assertions generated by bounded symbolic execution. Across P0-P4, the concrete value of `i` changes from `0` to a longer arithmetic expression, and the value of `max` may differ across branch-specific states. However, the memory skeleton is stable: each representative assertion contains permissions for `max`, `i`, `n`, and `a`, together with the full array predicate `IntArray.full_shape(a@pre, n@pre)`.

MALICE selects a later loop-head assertion to perform abstraction because it exposes the general shape after several iterations. The selected assertion has the following memory part:

```
data_at(&max, int, x_192) *
data_at(&i, int, (((0 + 1) + 1) + 1) + 1)) *
data_at(&n, int, n@pre) *
data_at(&a, int*, a@pre) *
IntArray::full_shape(a@pre, n@pre)
```

According to Step 1, this assertion is reduced using three targets.

For the scalar-variable target, the permissions of `max` and `i` are kept as scalar cells:

```
data_at(&max, int, x_192) *
data_at(&i, int, (((0 + 1) + 1) + 1) + 1))
```

The permissions of parameters `n` and `a` are also preserved:

```
data_at(&n, int, n@pre) *
data_at(&a, int*, a@pre)
```

For the processed-array target, MALICE abstracts the prefix from index `0` to the current loop index. This segment contains the already visited elements and excludes the current index:

```
IntArray::ceil_shape(a@pre, 0, i)
```

For the unprocessed-array target, MALICE abstracts the suffix from the current loop index to the original array length. This segment contains the current element `a[i]` and the elements that may be visited in later iterations:

```
IntArray::ceil_shape(a@pre, i, n@pre)
```

Thus the full-array predicate is reduced as follows:

```
IntArray::ceil_shape(a@pre, 0, i) *
IntArray::ceil_shape(a@pre, i, n@pre)
```

Combining the three targets gives the following intermediate memory assertion:

```
data_at(&max, int, x_192) *
data_at(&i, int, (((0 + 1) + 1) + 1) + 1)) *
data_at(&n, int, n@pre) *
data_at(&a, int*, a@pre) *
IntArray::ceil_shape(a@pre, 0, i) *
IntArray::ceil_shape(a@pre, i, n@pre)
```

This intermediate assertion still contains iteration-specific values. To make it usable as a loop invariant, MALICE generalizes these values into stable logical variables. The scalar value stored in `max` is represented by `max_v`, and the scalar value stored in `i` is represented by `i_v`. After this normalization, all array segment boundaries are expressed using the same loop-index abstraction `i_v`.

MALICE also checks whether any program-variable permission should be removed or converted to `undef_data_at`. In this example, no permission is removed: `max` is initialized before the loop, `i` is initialized in the loop header by `i = 0`, and `a` and `n` are function parameters.

**Output.** The generated memory assertion passed to Step 3 is:

```
data_at(&max, int, max_v) *
data_at(&i, int, i_v) *
data_at(&n, int, n@pre) *
data_at(&a, int*, a@pre) *
IntArray::ceil_shape(a@pre, 0, i_v) *
IntArray::ceil_shape(a@pre, i_v, n@pre)
```

### B.3.3. STEP 3: PURE CONSTRAINT GENERATION

**Input.** This step takes the memory assertion from Step 2 and the full set of loop-head assertions produced by bounded unrolling. Although only one representative state at each depth is displayed above, validation is performed against all branch-specific states generated at the checked depths.

**Analysis Chain.** MALICE generates candidate pure constraints and then validates them on the loop-unrolling information. A candidate is retained only if it holds in every checked loop-head state and contains only allowed pure-logic constructs.

The first candidate concerns the loop index:

```
0 <= i_v && i_v <= n@pre
```

This candidate is suggested by the loop initialization `i = 0`, the update `++i`, and the guard `i < n`. It is also required by the memory assertion, because `IntArray::ceil_shape(a@pre, 0, i_v)` and `IntArray::ceil_shape(a@pre, i_v, n@pre)` are meaningful only when the split point lies between the two array boundaries. The loop-unrolling information validates the candidate: P0 has `i = 0`; P1, P2, P3, and P4 show the successive values `0 + 1`, `(0 + 1) + 1`, `((0 + 1) + 1) + 1`, and `(((0 + 1) + 1) + 1) + 1`, together with the corresponding guards such as `0 < n@pre`, `(0 + 1) < n@pre`, and so on. The upper bound is non-strict because the invariant must also hold when the loop exits, where the index may equal `n@pre`.

The second candidate preserves the precondition on the array length:

```
0 <= n@pre && n@pre < INT_MAX
```

This candidate is directly present in P0 and remains present in all checked loop-head assertions. Since the loop does not modify n, the precondition-derived bounds on `n@pre` are stable.

The third candidate concerns the scalar variable `max`:

```
max_v >= -1
```

This candidate is suggested by the initialization `max = -1` and by the guarded update `if (a[i] > max) max = a[i]`. The loop-unrolling information validates the candidate across branch-specific states. At P0, `max` is exactly `-1`. At later depths, each state either keeps the previous value of `max` under a condition such as `x_111 <= -1`, or updates `max` under a condition such as `x_111 > -1` or `x_168 > x_144`. Thus the generated states support the lower bound `max_v >= -1`.

After validating the candidates, MALICE combines them with the memory assertion from Step 2.

**Output.** The generated complete loop invariant is:

```
/*@ Inv
∃ max_v i_v,
0 <= n@pre && n@pre < INT_MAX &&
0 <= i_v && i_v <= n@pre &&
max_v >= -1 &&
data_at(&max, int, max_v) *
data_at(&i, int, i_v) *
data_at(&n, int, n@pre) *
data_at(&a, int*, a@pre) *
IntArray::ceil_shape(a@pre, 0, i_v) *
IntArray::ceil_shape(a@pre, i_v, n@pre)
*/
```

This output is inserted into the invariant hole of the input program. The example illustrates that MALICE's intermediate analysis chains are also generated outputs: Step 1 outputs the predicate-target mapping, Step 2 outputs the reduction chain and memory assertion, and Step 3 outputs the candidate validation chain and the final invariant.

## B.4. Refinement

For simplicity, we only show one iteration of the refinement stage.

### B.4.1. INPUT

The input consists of the following parts:

- The source program with the candidate invariant.
- The feedback from the symbolic engine. Here,

```
fatal error: Entailment Check Failed:29:3
    .....(omitted).....
    13:   /*@ Inv Assert
    14: ∃ max_v i_v,
    15: 0 <= i_v && i_v <= n@pre &&
    16: 0 <= n@pre && n@pre < INT_MAX &&
    17: max_v >= -1 &&
    18: data_at(&max, int, max_v) *
    19: data_at(&i, int, i_v) *
    20: data_at(&n, int, n@pre) *
    21: data_at(&a, int*, a@pre) *
    22: IntArray::ceil_shape(a@pre, 0, i_v) *
    23: IntArray::ceil_shape(a@pre, i_v, n@pre)
    24: */
    25:   for (i = 0; i < n; ++i) {
    26:     if (a[i] > max) {
    27:       max = a[i];
    28:     }
>>> 29:   }
          ^~~~ ERROR HERE
    30:   return max;
    31: }
```

```
Interpret to error position:
Current Assertions:

@nrm
unnamed:
-------Assertion begin---------
branch name : NULL
next branch name : NULL
spec name : NULL
∃ max_v_90 i_v_91 v1_96 v1_100 ,
(v1_96 > max_v_90) &&
(i_v_91 < n@pre) &&
( 0  <= i_v_91) &&
(i_v_91 <= n@pre) &&
( 0  <= n@pre) &&
(n@pre < INT_MAX) &&
IntArray.ceil_shape (a@pre, i_v_91, n@pre) *
data_at ( (&max) , signed int , v1_100 ) *
data_at ( (&i) , signed int , i_v_91 ) *
data_at ( (&n) , signed int , n@pre ) *
data_at ( (&a) , signed int* , a@pre ) *
IntArray.ceil_shape (a@pre,  0 , i_v_91)
-------Assertion end ----------

@brk
NULL

@cnt
NULL

@ret
NULL
```

- The symbolic execution trace leading to the failure, given by the symbolic engine. Here, (for simplicity only a few tail states are shown)

```
==== Symbolic Execution into function array_max ====
...(omitted)...

-------Assertion begin---------
branch name : NULL
next branch name : NULL
spec name : NULL
∃ max_v_90 i_v_91 v1_96 ,
(v1_96 <= max_v_90) &&
(i_v_91 < n@pre) &&
( 0  <= i_v_91) &&
(i_v_91 <= n@pre) &&
( 0  <= n@pre) &&
(n@pre < INT_MAX) &&
(max_v_90 >= (- 1 )) &&
IntArray.ceil_shape (a@pre, i_v_91, n@pre) *
data_at ( (&max) , signed int , max_v_90 ) *
data_at ( (&i) , signed int , i_v_91 ) *
data_at ( (&n) , signed int , n@pre ) *
data_at ( (&a) , signed int* , a@pre ) *
IntArray.ceil_shape (a@pre,  0 , i_v_91)
-------Assertion end ----------
},
"Assertion␣break": {
},
"Assertion␣continue": {
}
</Asrt Tuple>

<Code Content>
}
</Code Content>

<Asrt Tuple>
"Assertion␣normal": {
-------Assertion begin---------
branch name : NULL
next branch name : NULL
spec name : NULL
∃ max_v_90 i_v_91 v1_96 v1_100 ,
```

```
(v1_96 > max_v_90) &&
(i_v_91 < n@pre) &&
( 0  <= i_v_91) &&
(i_v_91 <= n@pre) &&
( 0  <= n@pre) &&
(n@pre < INT_MAX) &&
IntArray.ceil_shape (a@pre, i_v_91, n@pre) *
data_at ( (&max) , signed int , v1_100 ) *
data_at ( (&i) , signed int , i_v_91 ) *
data_at ( (&n) , signed int , n@pre ) *
data_at ( (&a) , signed int* , a@pre ) *
IntArray.ceil_shape (a@pre,  0 , i_v_91)
-------Assertion end ----------
-------Assertion begin---------
branch name : NULL
next branch name : NULL
spec name : NULL
∃ max_v_90 i_v_91 v1_96 ,
(v1_96 <= max_v_90) &&
(i_v_91 < n@pre) &&
( 0  <= i_v_91) &&
(i_v_91 <= n@pre) &&
( 0  <= n@pre) &&
(n@pre < INT_MAX) &&
(max_v_90 >= (- 1 )) &&
IntArray.ceil_shape (a@pre, i_v_91, n@pre) *
data_at ( (&max) , signed int , max_v_90 ) *
data_at ( (&i) , signed int , i_v_91 ) *
data_at ( (&n) , signed int , n@pre ) *
data_at ( (&a) , signed int* , a@pre ) *
IntArray.ceil_shape (a@pre,  0 , i_v_91)
-------Assertion end ----------
},
"Assertion␣break": {
},
"Assertion␣continue": {
}
</Asrt Tuple>
Check <for> loop invariant here.
---------------Entailment----------------------------
Pre:
-------Assertion begin---------
branch name : NULL
next branch name : NULL
spec name : NULL
∃ max_v_90 i_v_91 v1_96 ,
(v1_96 <= max_v_90) &&
(i_v_91 < n@pre) &&
( 0  <= i_v_91) &&
(i_v_91 <= n@pre) &&
( 0  <= n@pre) &&
(n@pre < INT_MAX) &&
(max_v_90 >= (- 1 )) &&
IntArray.ceil_shape (a@pre, i_v_91, n@pre) *
data_at ( (&max) , signed int , max_v_90 ) *
data_at ( (&i) , signed int , (i_v_91 +  1 ) ) ) *
data_at ( (&n) , signed int , n@pre ) *
data_at ( (&a) , signed int* , a@pre ) *
IntArray.ceil_shape (a@pre,  0 , i_v_91)
-------Assertion end ----------
-------Assertion begin---------
branch name : NULL
next branch name : NULL
spec name : NULL
∃ max_v_90 i_v_91 v1_96 v1_100 ,
(v1_96 > max_v_90) &&
(i_v_91 < n@pre) &&
( 0  <= i_v_91) &&
(i_v_91 <= n@pre) &&
( 0  <= n@pre) &&
(n@pre < INT_MAX) &&
IntArray.ceil_shape (a@pre, i_v_91, n@pre) *
data_at ( (&max) , signed int , v1_100 ) *
data_at ( (&i) , signed int , (i_v_91 +  1 ) ) ) *
data_at ( (&n) , signed int , n@pre ) *
data_at ( (&a) , signed int* , a@pre ) *
IntArray.ceil_shape (a@pre,  0 , i_v_91)
-------Assertion end ----------
→
Post:
-------Assertion begin---------
```

```
branch name : NULL
next branch name : NULL
spec name : NULL
∃ max_v_88 i_v_89 ,
( 0   <= i_v_89) &&
(i_v_89 <= n@pre) &&
( 0   <= n@pre) &&
(n@pre < INT_MAX) &&
(max_v_88 >= (- 1 )) &&
data_at ( (&max) , signed int , max_v_88 ) *
data_at ( (&i) , signed int , i_v_89 ) *
data_at ( (&n) , signed int , n@pre ) *
data_at ( (&a) , signed int* , a@pre ) *
IntArray.ceil_shape (a@pre,  0 , i_v_89) *
IntArray.ceil_shape (a@pre, i_v_89, n@pre)
-------Assertion end ----------
--------------------------------------------------
Branch 1 result : Prop List can not be entailed
Res Prop:
(v1_100 >= (- 1 ))
Failed Entailment:
branch name : NULL
next branch name : NULL
spec name : NULL
∃ max_v_90 i_v_91 v1_96 ,
(v1_96 > max_v_90) &&
(i_v_91 < n@pre) &&
( 0   <= i_v_91) &&
(i_v_91 <= n@pre) &&
( 0   <= n@pre) &&
(n@pre < INT_MAX) &&
IntArray.ceil_shape (a@pre, i_v_91, n@pre) *
IntArray.ceil_shape (a@pre,  0 , i_v_91)

==>
branch name : NULL
next branch name : NULL
spec name : NULL
∃ i_v_91 ,
( 0   <= (i_v_91 +  1 )) &&
((i_v_91 +  1 ) <= n@pre) &&
( 0   <= n@pre) &&
(n@pre < INT_MAX) &&
IntArray.ceil_shape (a@pre,  0 , (i_v_91 +  1 )) *
IntArray.ceil_shape (a@pre, (i_v_91 +  1 ), n@pre)

Res Prop:
(v1_100 >= (- 1 ))

FAILED
=================================================================================================
```

- Few-shot examples of how to first, interpret the error message and diagnose the failure, then, propose a refined invariant. For simplicity, we omit here and the full version is available in our code.

### B.4.2. OUTPUT

We show a brief representative refinement output. The model first interprets the verification failure and then proposes a revised invariant.

**Error Interpretation and Diagnosis Step**

```
stderr.txt中显示Entailment Check Failed，根据ERROR HERE标识符指向for循环的右括号'}'。这个位置报错代表:
**循环不变量 => 符号执行一次循环体之后的循环不变量**
这个蕴含关系发生了错误，循环不变量没有保持不变。

对于蕴含关系错误，stdout.txt里面最后几段的信息会很有用，他会将工具未能推导出的entailment，以及其自然语言的解释打印出来。

从stdout.txt中可以看到:
```
Failed Entailment:
exists max_v_90 i_v_91 v1_96,
(v1_96 > max_v_90) &&
(i_v_91 < n@pre) &&
( 0   <= i_v_91) &&
(i_v_91 <= n@pre) &&
( 0   <= n@pre) &&
```

```
(n@pre < INT_MAX) &&
IntArray.ceil_shape (a@pre, i_v_91, n@pre) *
IntArray.ceil_shape (a@pre,  0, i_v_91)

==>
exists i_v_91,
( 0  <= (i_v_91 +  1 )) &&
((i_v_91 +  1 ) <= n@pre) &&
( 0  <= n@pre) &&
(n@pre < INT_MAX) &&
IntArray.ceil_shape (a@pre,  0, (i_v_91 +  1 )) *
IntArray.ceil_shape (a@pre, (i_v_91 +  1 ), n@pre)

Res Prop:
(v1_100 >= (- 1 ))
```

这个蕴含关系要求从左侧推导出右侧。左侧有两个分离的断言：
1. `IntArray::ceil_shape(a@pre, i_v_91, n@pre)` – 表示数组a从i_v_91到n@pre-1的部分
2. `IntArray::ceil_shape(a@pre, 0, i_v_91)` – 表示数组a从0到i_v_91-1的部分

右侧需要：
1. `IntArray::ceil_shape(a@pre, 0, i_v_91 + 1)` – 表示数组a从0到i_v_91的部分
2. `IntArray::ceil_shape(a@pre, i_v_91 + 1, n@pre)` – 表示数组a从i_v_91+1到n@pre-1的部分

问题在于：左侧的两个`ceil_shape`断言是分离的，但右侧需要将它们合并为`ceil_shape(a@pre, 0, i_v_91 + 1)`。这需要知道`ceil_shape`的定义：`ceil_shape(a, start, end)`表示数组a从start到end-1的部分。

实际上，左侧的`IntArray::ceil_shape(a@pre, i_v_91, n@pre)`加上`IntArray::ceil_shape(a@pre, 0, i_v_91)`应该等价于`IntArray::ceil_shape(a@pre, 0, n@pre)`，而不是`IntArray::ceil_shape(a@pre, 0, i_v_91 + 1)`。

查看循环不变量：
```c
IntArray::ceil_shape(a@pre, 0, i_v) *
IntArray::ceil_shape(a@pre, i_v, n@pre)
```

这里有一个错误：`IntArray::ceil_shape(a@pre, i_v, n@pre)`表示从i_v到n@pre-1的部分，但根据`ceil_shape`的定义，它应该是从i_v到n@pre-1的部分。然而，循环不变量应该描述的是：已经处理过的部分（0到i_v-1）和尚未处理的部分（i_v到n@pre-1）。

实际上，`IntArray::full_shape(a@pre, n@pre)`应该被拆分为`IntArray::ceil_shape(a@pre, 0, i_v)`和`IntArray::ceil_shape(a@pre, i_v, n@pre)`，但这里有一个问题：`IntArray::ceil_shape(a@pre, i_v, n@pre)`表示从i_v到n@pre-1的部分，而`IntArray::ceil_shape(a@pre, 0, i_v)`表示从0到i_v-1的部分。这两部分合起来应该覆盖整个数组。

但问题在于：当i_v = n@pre时，`IntArray::ceil_shape(a@pre, i_v, n@pre)`表示从n@pre到n@pre-1的部分，即空部分，这是正确的。然而，当i_v < n@pre时，我们需要能够从两个部分合并出`IntArray::ceil_shape(a@pre, 0, n@pre)`。

实际上，循环不变量应该使用`IntArray::ceil_shape(a@pre, 0, i_v)`表示已处理部分，`IntArray::ceil_shape(a@pre, i_v, n@pre)`表示未处理部分。但这里缺少了一个关键信息：`IntArray::ceil_shape(a@pre, 0, i_v) * IntArray::ceil_shape(a@pre, i_v, n@pre)`应该等价于`IntArray::full_shape(a@pre, n@pre)`。

从错误信息看，工具无法自动推导这个等价关系。我们需要在循环不变量中明确这个关系，或者使用不同的断言。

## Refined Invariant Proposal Step

根据错误解释，问题在于循环不变量中的数组断言无法正确推导。我们需要修改循环不变量，使其更准确地描述数组状态。

正确的循环不变量应该是：
1. `IntArray::ceil_shape(a@pre, 0, i_v)` – 已处理部分（0到i_v-1）
2. `IntArray::ceil_shape(a@pre, i_v, n@pre)` – 未处理部分（i_v到n@pre-1）

但为了能够推导出循环结束后的后条件，我们需要知道整个数组的完整权限。实际上，循环不变量应该保持`IntArray::full_shape(a@pre, n@pre)`，因为数组在循环中没有被修改。

因此，正确的循环不变量应该是：
```c
IntArray::full_shape(a@pre, n@pre)
```

但这样无法跟踪循环进度。我们需要一个更精确的表示：数组的完整权限被拆分为已处理部分和未处理部分。

实际上，`IntArray::full_shape(a@pre, n@pre)`可以拆分为：
```c
IntArray::ceil_shape(a@pre, 0, i_v) * IntArray::ceil_shape(a@pre, i_v, n@pre)
```

但工具无法自动推导这个拆分。我们需要在循环不变量中明确这个关系，或者使用不同的断言。

查看其他类似案例（如array_copy1），正确的做法是使用`IntArray::ceil_shape`来表示数组的部分。问题可能在于我们需要确保`IntArray::ceil_shape`的正确使用。

实际上，从错误信息看，问题可能在于`IntArray::ceil_shape(a@pre, i_v, n@pre)`的定义。我们需要检查`ceil_shape`的定义是否正确。

根据常见的定义，`IntArray::ceil_shape(a, start, end)`表示数组a从start到end-1的部分。因此：
- `IntArray::ceil_shape(a@pre, 0, i_v)` 表示从0到i_v-1的部分
- `IntArray::ceil_shape(a@pre, i_v, n@pre)` 表示从i_v到n@pre-1的部分

这两部分合起来应该覆盖整个数组。但工具无法自动推导这个关系。

我们需要修改循环不变量，使其更简单。由于数组在循环中没有被修改，我们可以使用：
```c
IntArray::full_shape(a@pre, n@pre)
```

但这样无法跟踪循环进度。我们需要一个折中方案：使用`IntArray::full_shape(a@pre, n@pre)`，但添加一个条件来跟踪循环进度。

实际上，循环不变量可以这样写：
```c
IntArray::full_shape(a@pre, n@pre) &&
0 <= i_v && i_v <= n@pre
```

这样既保持了数组的完整权限，又跟踪了循环进度。

让我们修改循环不变量为：
```c
IntArray::full_shape(a@pre, n@pre) &&
0 <= i_v && i_v <= n@pre
```

这样，循环不变量就能保持不变，因为数组在循环中没有被修改。

### 完整修正代码
```c
#include "verification_stdlib.h"
#include "verification_list.h"
#include "int_array_def.h"
#include "sll_shape_def.h"

int array_max(int *a, int n)
/*@ Require 0 <= n && n < INT_MAX && IntArray::full_shape(a, n)
    Ensure IntArray::full_shape(a, n)
*/
{
  int i;
  int max = -1;
  /*@ Inv Assert
exists max_v i_v,
0 <= i_v && i_v <= n@pre &&
0 <= n@pre && n@pre < INT_MAX &&
max_v >= -1 &&
data_at(&max, int, max_v) *
data_at(&i, int, i_v) *
data_at(&n, int, n@pre) *
data_at(&a, int*, a@pre) *
IntArray::full_shape(a@pre, n@pre)
*/
  for (i = 0; i < n; ++i) {
    if (a[i] > max) {
      max = a[i];
    }
  }
  return max;
}
```

## C. Configuration of SFT

Our main SFT configuration is as follows:

- model_name_or_path: unsloth/Meta-Llama-3.1-8B-Instruct
- stage: sft
- do_train: true
- finetuning_type: lora
- lora_rank: 16

- `lora_target:` `all`
- `per_device_train_batch_size:` 1
- `gradient_accumulation_steps:` 8
- `learning_rate:` 2e-4
- `weight_decay:` 0.01
- `num_train_epochs:` 3.0
- `lr_scheduler_type:` `cosine`
- `warmup_ratio:` 0.1
- `bf16:` `true`
- `ddp_timeout:` 180000000

