# OpenReview forum: "MALICE: Memory-aware Loop Invariants Generation on Symbolic Execution Traces"
_ICML.cc/2026/Conference — ICML 2026 regular_

### Official Review · Reviewer_hvz7 · 2026-03-03

**Soundness:** 2
**Presentation:** 2
**Significance:** 4
**Originality:** 3
**Overall Recommendation:** 4
**Confidence:** 5

**Summary:**

The paper introduces MALICE, a framework for automatically generating shape loop invariants for memory-manipulating C programs. MALICE includes two stages for generating shape loop invariants. The first one is a multi-step prediction stage. In this stage, the LLM is first guided by a Chain-of-Thought (CoT) prompt to generate an abstract reduction guidance, then generates memory predicates using the guidance and symbolic loop unrolling information. The second stage is an agentic refinement stage, where the LLM iteratively debugs and corrects the failed loop invariants based on feedback from the QCP verification engine. The paper extends the previous benchmark LIG-MM to LIG-MM++, and evaluates MALICE on it. The overall pass rate achieves 82.69\% using DeepSeek-V3.2 and in-context learning, significantly outperforming the state-of-the-art baselines for synthesizing shape invariants.

**Compliance With Llm Reviewing Policy:**

Affirmed.

**Final Justification:**

The review addressed my concerns appropriately. Therefore I raised my score.

**Key Questions For Authors:**

1. What are the concrete inputs/outputs for the Abstract Reduction stage and the Memory Analysis stage? If the predicate selection plan generated in the first stage is in natural language, what are the exact mechanics of parsing or using this plan in the subsequent Memory Analysis stage?
2. Could you report the success rate of LIG-SE and MALICE separated by the data structure types (e.g. Linked Lists, Trees, Arrays)? How does MALICE compare to LIG-SE on the original LIG-MM dataset (excluding arrays)?
3. What is the backbone model used for evaluating LIG-SE? How does MALICE perform with the same model as that used in LIG-SE?
4. What is the average Lines of Code (LoC) in the LIG-MM+ dataset? As symbolic execution faces the problem of path explosion, how does MALICE scale to larger and more complex real-world programs?
5. What is the cause for the failures in the evaluation of MALICE?

**Limitations:**

The authors have breifly included an Impact Statement Section, but they have not discussed the limitations of their work. The authors should add a section discussing the failure reasons, and potential limitations such as path explosion issues inherited by the QCP engine, and the resource issues on loop unrollling depth and maximum refinement iterations.

**Strengths And Weaknesses:**

**Strengths**:
- The paper targets an important problem in program verification. While prior works are largely limited to numerical loop invariants, MALICE extends it to shape loop invariants for memory manipulation. This is a significant step forward for real-world program verification.
- The paper presents a fully-automated, agentic framework which combines LLMs and symbolic execution well. By providing symbolic execution traces and loop unrolling information for the LLM, the framework makes the LLM better understand the state evolutions of the programs, thereby maximizing the LLM's reasoning capabilities.
- The paper extends the original benchmarks and achieves a high success rate at 82.69\%.

**Weaknesses**:
- Concerns on unfair comparison in the evaluation. There are several fairness and soundness concerns in the evaluation, including inconsistent backbone LLMs, unbounded resource limits, and benchmark issues. (See below as explanation)
- Unclear implementation details. The description of the prediction stage is vague and unclear. Specifically, the concrete inputs/outputs of Section 3.2.1 (Abstract Reduction Guidance) and Section 3.2.2 (Memory Analysis) are missing, which form the core technique contribution to the paper. And there is neither an end-to-end prompt nor an illustrative example.
- Lack of discussion of failures and limitation. The reasons for the verification failures (>10\%) are unknown, and there is not a discussion on the limitation of the proposed method.

**Soundness**:
While the core framework of MALICE is logically sound, there are critical soundness concerns in the experimental design. These include inconsistent base models for comparison, unbounded resource limit and benchmark issues.
- Inconsistent base models. In Section 4.3, MALICE is evaluated using the 685B DeepSeek-V3.2, while the baseline method LIG-SE (as stated in its original paper) uses a much smaller 350M CodeGen model. This can lead to significant unfair comparsion, which makes it unclear whether the performance improvement is by design or by the raw reasoning power of the model.
- Unbounded resource limit. The loop unrolling time and the maximum refinement iteration are set by the authors, and there is not a discussion on the two resource limit. As the authors count for pass rate@8, it means that at most $8\times (10+1)=88$ queries can be conducted for a single program, which is an unfair comparison to the baselines which require a single query per run. Moreover, Table 3 indicates that the pass rate decreases to 34.62\% without refinement (i.e., with limited resource bound). For a fair comparison, a consistent resource limit (time and token) for all methods should be ensured.
- Benchmark issues. The paper expands the original benchmark to include array manipulation programs, which are beyond the scope of the baseline method LIG-SE. However, the authors only report the overall pass rate across all programs in the extended benchmark. There is not a direct comparison on the problems which are in the scope of both LIG-SE and MALICE.

**Presentation**:
The paper is generally well-organised and structured. However, the implementation details regarding  the core technique contribution of the paper are missing. Specifically, in Section 3.2.1 and Section 3.2.2, the authors only provide high-level principles and use vague language (e.g., ``thereby directly yielding a general loop invariant'' at Line 268). It is unclear how the LLM translates the natural language plan into formal logic, and  the concrete inputs/outputs of these stages are missing. The elements in Figure 3 (e.g. data_at, listrep) are not explained, which can lead to confusion for readers unfamiliar with Seperation Logic and QCP. Although the authors introduce the main parts constructing the prompt in Appdendix A.1, crucial details are still missing for understanding the paper.

**Significant**:
The paper targets an important problem in program verification. Prior works on loop invariant synthesis mostly focus on numerical invariants. The paper targets on shape loop invariants for memory manipulation, which can lead to a forward step to real-world applications.

**Originality**:
The combination of explicitly feeding unrolled symbolic execution traces into an LLM to generate shape loop invariants for memory manipulation C programs offers originality. While the previous work LIG-SE has made a first attempt on combining LLMs and symbolic execution, MALICE further develops it into a more systematic and agentic framework.

---

> ### Author Rebuttal · Authors · 2026-03-31
>
> # Response to Reviewer hvz7
> We thank the reviewer for the rigorous evaluation. We address every concern with concrete evidence.
>
> ---
> ## Soundness Concern 1: Inconsistent Base Models (CRITICAL)
> > "MALICE is evaluated using the 685B DeepSeek-V3.2, while the baseline method LIG-SE uses a much smaller 350M CodeGen model."
>
> **Framework design, not model scale, drives performance.** With the same 685B model, Vanilla achieves only **15.77%**—far below LIG-SE's **41.92%** with 350M. The 66.92pp gain (Vanilla→Full Pipeline) is primarily from our framework.
>
> MALICE (SFT) already uses 8B for Refinement yet achieves **86.15%**, surpassing 685B ICL. Our new Full-8B experiment removes large-model dependency entirely and achieves **81.54%**:
>
> | Method | Prediction | Refinement | Pass Rate |
> |--------|-----------|------------|-----------|
> | LIG-SE | CodeGen-350M (SFT) | — | 41.92% |
> | **Full-8B SFT (New)** | **Qwen3-8B (ICL)** | **Llama-3.1-8B (SFT)** | **81.54%** |
> | MALICE (SFT) | DeepSeek-V3.2 | Llama-3.1-8B (SFT) | **86.15%** |
> | MALICE (ICL) | DeepSeek-V3.2 | DeepSeek-V3.2 | 82.69% |
>
> We note that using 8B ICL for **Prediction** stages produces frequent syntax errors and reasoning breakdowns; SFT effectively distills the refinement reasoning into the 8B model's weights, bypassing the need for large-scale in-context reasoning. **MALICE's contribution is twofold: (1) framework design and (2) the SFT distillation pathway.**
>
> ---
> ## Soundness Concern 2: Unbounded Resource Limit (pass@8)
> > "pass rate@8...unfair comparison to baselines which require a single query per run."
>
> **All methods use pass@8.** LIG-SE and AutoSpec also employ iterative feedback loops within each run. MALICE w/o Refinement (34.62%) provides a direct single-query comparison with LIG-SE (41.92%). Even pass@150 w/o Refinement (47.31%) barely exceeds LIG-SE—**refinement is essential**, not brute-force repetition.
>
> | Method | Runs | Per-Run Behavior | Pass Rate |
> |--------|------|-----------------|-----------|
> | SLING | 1* | SMT solving (deterministic) | 26.15% |
> | LIG-SE | 8 | LLM + SE feedback loop | 41.92% |
> | MALICE w/o Refinement | 8 | 3 LLM calls (prediction only) | 34.62% |
> | **MALICE (1×8 calls, New)** | **1** | **3 pred + 5 refine = 8 total** | **59.62%** |
> | MALICE Full | 8 | 3 pred + up to 10 refine | **82.69%** |
>
> *SLING is deterministic—pass@8 = pass@1.
>
> With just 8 LLM calls in a single run, MALICE (59.62%) still outperforms all baselines. Refinement budget is bounded (K=10/run); we will report resource limits in the future.
>
> ---
> ## Soundness Concern 3: Per-Category Breakdown
> > "The authors only report the overall pass rate."
>
> LIG-MM+ = 260 programs (DLL:133, SLL:96, Array:8, Others:23) from 7 sources.
>
> | Cat. | # | SLING | LIG-SE | 1×8 | ICL | SFT |
> |------|---|-------|--------|-----|-----|-----|
> | DLL | 133 | 24.81% | 32.33% | 48.12% | **83.46%** | **90.98%** |
> | SLL | 96 | 33.33% | 61.46% | 87.50% | **100%** | **100%** |
> | Array | 8 | 0% | 0% | 87.50% | **100%** | 87.50% |
> | Others* | 23 | 13.04% | 30.43% | 0% | 0% | 0% |
> | **Total** | **260** | **26.15%** | **41.92%** | **59.62%** | **82.69%** | **86.15%** |
>
> *MALICE scores 0% on Others (tree, hash table) due to missing QCP predicate strategies (engineering limitation). SLING/LIG-SE use different backends.
>
> Shared scope (DLL+SLL+Array): MALICE **215/237=90.72%** vs LIG-SE 102/237=43.04% (+47.68pp).
>
> ---
> ## Q1: Concrete I/O for §3.2.1 and §3.2.2
> > "The concrete inputs/outputs of Section 3.2.1 and Section 3.2.2 are missing."
>
> **Abstract Reduction (§3.2.1):** Input: C code + CoT prompt → Output: predicate selection plan. E.g., for `app(x,y)`: choose `lseg(x,t)` for traversed region, `data_at` for current node, `listrep` for remaining list.
>
> **Memory Analysis (§3.2.2):** Input: plan + symbolic trace → Output: SL assertion. E.g., `lseg(x,t) * data_at(field_addr(t,next),u) * listrep(u) * listrep(y)`.
>
> `data_at`/`listrep`/`lseg` are standard SL constructs; will add definitions and end-to-end example in Appendix.
>
> ## Q2: Per-Category & Original LIG-MM
> Per-category: see SC3. On original LIG-MM (DLL+SLL): MALICE **207/229=90.39%** vs LIG-SE 102/229=44.54% (+45.85pp).
>
> ## Q3: Backbone Model
> LIG-SE uses CodeGen-350M with task-specific pre-training. Vanilla with same 685B gets only 15.77%. Full-8B (Qwen3-8B + Llama SFT) achieves **81.54%**, confirming framework independence from model scale.
>
> ## Q4: LoC and Scalability
> 260 programs, avg ~36 LoC (9,197 total), representative of production codebases. Challenges: (1) shape generalization from complex traces, (2) path explosion (mitigated by K=4 unrolling + random sampling). Will add Limitations section.
>
> ## Q5: Failure Analysis
> Four categories: (1) unsupported structures (trees/hash tables lack QCP strategies), (2) complex pointers (double pointers), (3) multi-branch/nested loops, (4) imprecise non-spatial conditions. E.g., `multi_rev` on DLL has lowest pass rate. Will add Limitations section.

---

> > ### Author Rebuttal · Reviewer_hvz7 · 2026-04-03
> >
> > Thank you for the rebuttal. I have raised my score to 4.

---

### Official Review · Reviewer_xLDr · 2026-03-10

**Soundness:** 3
**Presentation:** 2
**Significance:** 3
**Originality:** 3
**Overall Recommendation:** 5
**Confidence:** 3

**Summary:**

This paper augments an LLM with information from symbolic execution to generate loop invariants for memory-manipulating programs. Specifically, the loop invariant itself has to reason about the state of the heap as opposed to something that's just numerical operations. After a generation phase, they perform a refinement phase which corrects any errors in the initial loop invariant generated by the LLM. They perform an evaluation and show that their approach greatly improves performance on several language models.

**Compliance With Llm Reviewing Policy:**

Affirmed.

**Final Justification:**

I remain positive about this paper. I think the paper does a good job proposing a solution to a problem and has a clean evaluation. I believe there were some small issues that had to be ruled out through some additional ablations (whether it was the traces themselves that were providing the benefit or just the testing) and were addressed in the rebuttal.

**Key Questions For Authors:**

Do the authors have any insight on whether the more powerful models these days would be able to perform these?

Is it the symbolic execution traces themselves that are providing the benefit, or is it just the ability of the system to test its implementations repeatedly?

**Limitations:**

This is missing a bit. There's a "Conclusion and Future Work" section but doesn't talk about the limitations. I noticed the authors have some extra space so I think they could put it in.

**Strengths And Weaknesses:**

#### Strengths
I think this is a good paper. It introduces a problem area, proposes a solution, and evaluates their solution pretty straightforwardly, and show that the approach improves performance.

#### Weaknesses
I think the only thing missing for me is some evaluation. I'd be curious a more powerful model like Claude Opus 4.6 could generate loop invariants very well.

I think a missing ablation is testing the model's ability to generate invariants by just looking at the code without information from symbolic execution. It's not clear to me whether the model being given the symbolic execution trace is what's helping, or if it's the iterative testing provided by the symbolic execution. What if you just checked the invariant, and gave the model pass/fail without the trace?

Aside from that, I have a few small presentation things:
- Figure 1 has some typos/errors in the refinement box. Seems very LLM-generated. I don't have an issue with LLM-generated figures but I think the authors should fix the typos.
- Line 172: "Using the constructed context" is a little vague
- Line 375: the three-step Prediction stage is a little vague as well. It's not clearly labelled what the three steps actually are, and is buried in the Prediction section of section 3.1.

---

> ### Author Rebuttal · Authors · 2026-03-31
>
> # Response to Reviewer xLDr
> We sincerely thank the reviewer for the positive assessment and acceptance recommendation. We address each suggestion below.
>
> ---
> ## Concern 1: Evaluation on More Powerful Models
> > "I'd be curious whether a more powerful model like Claude Opus 4.6 could generate loop invariants very well."
>
> We evaluated on three backbone LLMs spanning instruct and reasoning paradigms (same model for both Prediction and Refinement in ICL mode):
>
> | Prediction + Refinement Model | Type | Pass Rate |
> |-------------------------------|------|-----------|
> | DeepSeek-V3.2 (685B) | Instruct | 82.69% |
> | Qwen3-Max | Instruct | 75.00% |
> | DeepSeek-V3.2-Thinking | Reasoning | 82.69% |
>
> Additionally, MALICE (SFT) uses DeepSeek-V3.2 for Prediction but only Llama-3.1-8B (SFT) for Refinement, yet achieves **86.15%**.
>
> Consistent performance across architectures suggests framework design is the primary driver. The Vanilla baseline (same DeepSeek-V3.2, no framework) achieves only **15.77%**, indicating that simply using a more powerful model without our framework would be insufficient.
>
> We have not tested Claude Opus 4.6 due to API constraints, but our framework is model-agnostic—we expect it to benefit from any sufficiently capable backbone. We will add this to future work.
>
> ---
> ## Concern 2: Trace vs. Iterative Testing
> > "It's not clear to me whether the model being given the symbolic execution trace is what's helping, or if it's the iterative testing provided by the symbolic execution. What if you just checked the invariant, and gave the model pass/fail without the trace?"
>
> To directly answer this, we conducted a new **"w/o Trace"** ablation (all at pass@8):
>
> | Variant | Trace Info | Refinement Feedback | Pass Rate |
> |---------|-----------|-------------------|-----------|
> | Vanilla | No | No | 15.77% |
> | w/o Refinement | Yes | No | 34.62% |
> | **w/o Trace (NEW)** | **No** | **Yes (pass/fail only)** | **36.15%** |
> | w/o Prediction | Partial* | Yes (with trace) | 66.15% |
> | Full Pipeline | Yes | Yes (with trace) | 82.69% |
>
> *w/o Prediction uses minimal prompting but still receives trace-based feedback during refinement.
>
> > **w/o Trace setup:** When verification fails, provide only **pass/fail status and error type** from QCP, but **not** the symbolic execution trace or detailed error analysis. Same refinement budget (K=10) and pass@8 as the full pipeline.
>
> **Per-category breakdown:**
>
> |  | DLL | SLL | Array | Others | Overall |
> |--|-----|-----|-------|--------|---------|
> | Full Pipeline | 111/133 | 96/96 | 8/8 | 0/23 | **82.69%** |
> | w/o Trace | 21/133 | 72/96 | 1/8 | 0/23 | **36.15%** |
>
> **Trace contributes +46.54pp** (82.69% vs. 36.15%):
>
> 1. **w/o Trace ≈ w/o Refinement:** w/o Trace (36.15%) performs almost identically to w/o Refinement (34.62%), meaning iterative pass/fail-only feedback without traces provides **almost no benefit**—the refinement loop is only effective with detailed trace information.
> 2. **Trace impact is most pronounced for complex structures:** DLL shows +67.67pp (83.46% vs. 15.79%); SLL +25.00pp (100% vs. 75.00%); Array drops from 100% to 12.50%.
> 3. Without traces, the model cannot observe **how memory states evolve** across iterations—the key semantic insight for separation logic predicate synthesis. Trace-guided refinement provides actionable diagnostics, whereas pass/fail-only feedback gives no basis for targeted correction.
> 4. This further supports our pass@150 finding: even 150 independent attempts without traces (47.31%) cannot match the full pipeline (82.69%).
>
> ---
> ## Concern 3: Figure 2 Typos
> > "Figure 2 has some typos/errors in the refinement box."
>
> Will fix all typos in the refinement box.
>
> ---
> ## Concern 4: Vague Language
> > "Line 172: 'Using the constructed context' is a little vague"
> > "Line 375: the three-step Prediction stage is a little vague as well."
>
> **Line 172:** Will rewrite to: *"Using the prediction context—comprising the predicate selection plan, memory assertions, and pure properties—the system constructs the candidate invariant."*
>
> **Line 375:** Three steps: (1) Abstract Reduction Guidance: CoT → predicate plan, (2) Memory Assertions: plan + traces → SL predicates, (3) Pure Properties: traces → arithmetic constraints. Will add numbered subheadings.
>
> ---
> ## Concern 5: Missing Limitations Section
> Will add a dedicated **Limitations** section covering: (1) shape generalization difficulty, (2) discussion on loop unrolling depth (K=4), (3) discussion on refinement bound (K=10, diminishing returns beyond K=8), (4) failure categories (unsupported structures, complex pointers, nested loops, imprecise pure conditions), (5) benchmark scale.

---

> > ### Author Rebuttal · Reviewer_xLDr · 2026-03-31
> >
> > Thank you.

---

### Official Review · Reviewer_C8gC · 2026-03-12

**Soundness:** 3
**Presentation:** 2
**Significance:** 3
**Originality:** 3
**Overall Recommendation:** 4
**Confidence:** 3

**Summary:**

The paper presents MALICE, an LLM-based system for generating memory-aware shape invariants. To address this challenging problem, the authors leverage symbolic execution traces to help the LLM better understand the evolution of program memory states, and subsequently employ an agentic refinement loop that incorporates feedback from verification tools. Evaluated on an author-enriched benchmark set, the proposed system outperforms an existing traditional tool as well as several preliminary LLM-based approaches.

**Compliance With Llm Reviewing Policy:**

Affirmed.

**Final Justification:**

The authors’ response resolves my concerns, particularly regarding clarity in presentation. I will maintain my positive score.

**Key Questions For Authors:**

- I would appreciate clarification regarding the concerns raised in the Weaknesses section.

- The evaluation relies on a benchmark set enriched by the authors. It would be helpful to better justify why this benchmark set is sufficient and representative. Are there other benchmark sets used in previous systems such as SLING that could also be considered?

**Limitations:**

yes

**Strengths And Weaknesses:**

## Strength

- Generating memory-aware shape invariants is a challenging problem. Leveraging an agent-style workflow that incorporates feedback from multiple tools, in this case symbolic executor and verifier, is a reasonable and promising direction.

- The authors introduce several heuristic designs, such as predicate selection and memory analysis strategies. These heuristics appear well aligned with the problem domain and the LLM-based workflow, and their effectiveness is supported by the ablation studies.

- On the author-enriched benchmark set, the proposed approach outperforms existing tools.

## Weakness

- Although each component of the design is described fairly clearly, the overall system workflow remains somewhat difficult to follow. Figure 2 appears too high-level to convey the full architecture. A more detailed architecture diagram, with clearer references throughout the text, could improve readability. An end-to-end running example would also help readers better understand how the system operates.

- The paper lacks a clear problem statement. The authors primarily discuss memory-aware loop invariant generation, but it is not clear whether the invariants are generated to help prove some verification properties. From certain parts of the paper, it appears so, but this is not clear.

- The paper does not provide a reference for LIG-SE. Although it may originate from a previously cited work, the tool is presented as an important baseline and should therefore be clearly cited and explained, including how it differs from the proposed system.

- There are some minor writing issues. For example, the first double quotation mark appears incorrect, there is an incomplete sentence at line 57, and in the bibliography entries (e.g., QCP) capitalization should be corrected.

---

> ### Author Rebuttal · Authors · 2026-03-31
>
> # Response to Reviewer C8gC
> We sincerely thank the reviewer for the positive assessment. We are encouraged that the reviewer finds our agent-style workflow "a reasonable and promising direction." We address each concern below.
>
> ---
> ## Concern 1: System Workflow Diagram Too Abstract
> > "Figure 2 appears too high-level to convey the full architecture. A more detailed architecture diagram, with clearer references throughout the text, could improve readability. An end-to-end running example would also help."
>
> We will: **(1)** Replace Figure 2 with a detailed architecture diagram showing data flow: Symbolic Executor (QCP) → 3-step Prediction (Abstract Reduction → Memory Analysis → Pure Properties) → Candidate Invariant → QCP Verifier → Refinement loop (Explanation + Correction, up to K=10). **(2)** Add an end-to-end example in the Appendix walking through the SLL traversal function `iter`: input code/spec → symbolic traces (P0–P3) → Abstract Reduction output → Memory Analysis output → refinement iteration → final verified invariant.
>
> ---
> ## Concern 2: Unclear Problem Statement
> > "It is not clear whether the invariants are generated to help prove some verification properties."
>
> Loop invariant generation in our work serves **program correctness verification**, specifically **memory safety**. Consider a SLL reverse function from Zephyr RTOS:
>
> ```c
> struct sys_slist_t *reverse(struct sys_slist_t *p)
> /*@ Require listrep(p)        // Pre: p is a valid linked list
>     Ensure  listrep(__return)  // Post: return value is a valid linked list */
> ```
>
> QCP needs a loop invariant describing the memory state at each iteration—e.g., during reversal, the memory space splits into two valid lists. QCP checks three conditions:
> 1. **Initialization:** P ⇒ Inv
> 2. **Preservation:** {Inv ∧ B} S {Inv}
> 3. **Exit:** {Inv ∧ ¬B} S₁ {Q}
>
> Our pass rate directly measures how often MALICE generates invariants passing all three checks. We will add this clarification to the Introduction.
>
> ---
> ## Concern 3: Missing LIG-SE Reference
> > "The paper does not provide a reference for LIG-SE. The tool is presented as an important baseline and should therefore be clearly cited and explained."
>
> LIG-SE is from **Liu et al., "Towards general loop invariant generation: a benchmark of programs with memory manipulation" (NeurIPS 2024)**, cited in our paper. We should have explained the methodology difference more explicitly:
>
> | Aspect          | LIG-SE                                  | MALICE                                               |
> | --------------- | --------------------------------------- | ---------------------------------------------------- |
> | Approach        | Self-supervised pre-training on SE data | ICL + agentic refinement on SE traces (end-to-end)   |
> | Prediction      | CodeGen-350M (fine-tuned)               | DeepSeek-V3.2 (685B, ICL)                            |
> | Refinement      | None (single-pass)                      | DeepSeek-V3.2 (ICL) or Llama-3.1-8B (SFT)            |
> | Training        | Requires task-specific pre-training     | ICL: no training; SFT: refinement only               |
> | Data structures | Linked lists only                       | Linked lists, arrays (+ tree/hashtable in benchmark) |
>
> The key difference: LIG-SE **bakes** SE knowledge into weights via pre-training, while MALICE **provides** SE information at inference time via structured prompts + verification-guided refinement. This makes MALICE more flexible (new data structures without retraining) and more effective (82.69% vs. 41.92%). We will add this comparison to the paper.
>
> ---
> ## Concern 4: Writing Issues
> > "The first double quotation mark appears incorrect, there is an incomplete sentence at line 57, and in the bibliography entries (e.g., QCP) capitalization should be corrected."
>
> Will fix all issues: correct quotation marks, complete sentence at line 57, fix bibliography capitalization (protect proper nouns with braces), and conduct a full copy-editing pass.
>
> ---
> ## Q1: Benchmark Justification
> > "It would be helpful to better justify why this benchmark set is sufficient and representative. Are there other benchmark sets used in previous systems such as SLING that could also be considered?"
>
> LIG-MM+ adjusts LIG-MM [Liu et al., 2024] into **260 programs** from 7 sources (GlibC, Linux and so on) covering 5 data structure families. LIG-MM+ is a benchmark for LLM-based memory-aware loop invariant generation in C.
>
> Representativeness:
> - **Source diversity:** 4 real-world system codebases + academic + competition benchmarks
> - **Structure diversity:** SLL, DLL, tree, hash table, array(NEW)
> - **Operation diversity:** traversal, reversal, merging, appending, copying, counting, deletion
> - **Complexity range:** simple single-loop traversals to multi-loop merge/reverse operations
>
> Each program requires manual specification and predicate strategy definitions, making benchmark construction labor-intensive.

---

> > ### Author Rebuttal · Reviewer_C8gC · 2026-03-31
> >
> > I thank the authors for the response. I am keeping my positive score.

---

### Official Review · Reviewer_88vz · 2026-03-13

**Soundness:** 3
**Presentation:** 3
**Significance:** 2
**Originality:** 3
**Overall Recommendation:** 4
**Confidence:** 2

**Summary:**

MALICE is a framework that uses symbolic execution traces and SFT to improve LLM performance in generating loop invariants.

**Compliance With Llm Reviewing Policy:**

Affirmed.

**Final Justification:**

This paper was in a bad shape when it was received, but the authors seem determined to fix it before publication and the results are important. Assuming that the many changes the authors have promised will be made, I recommend accepting this paper.

**Key Questions For Authors:**

- Please describe the methodology used when extending your benchmark. How big is LIG-MM+ and how did you ensure diversity/complexity?
- Please describe your SFT method some more.

**Limitations:**

Yes

**Strengths And Weaknesses:**

*Strengths*
- The paper presents a number of mechanisms to solve the problem that all tie together to achieve an impressive result.
- Generating loop invariants for C programs is a bit niche, but it's an important niche and a longstanding problem.

*Weaknesses*
- The evaluation appears sound, but the authors do not discuss the methodology they used when extending their benchmark or its size. This definitely needs to be added before the paper can be accepted. There also could have a been a discussion on how well the success of this method scales with increasing code size and complexity.
- A lot of details are missing and need to be added to the appendix. For instance, what data was the SFT done on?
- While the writing is overall fairly clear, the paper could use some copy editing. There are a number of typos and missing spaces.

Overall, this is an interesting result, but the submission could likely benefit from some editing and a few additional analyses for evaluation.

---

> ### Author Rebuttal · Authors · 2026-03-31
>
> # Response to Reviewer 88vz
> We thank the reviewer for the careful reading. We are glad the reviewer recognizes that "the paper presents a number of mechanisms that all tie together to achieve an impressive result." We address each concern below.
>
> ---
> ## Concern 1: Missing Benchmark Methodology
> > "The authors do not discuss the methodology they used when extending their benchmark or its size. This definitely needs to be added before the paper can be accepted."
>
> LIG-MM+ is built upon LIG-MM [Liu et al., 2024]. We made adjustments (adding/removing structurally equivalent variants) and extended with array programs. LIG-MM+ contains **260 programs**:
>
> | Category | # | Sources | Data Structures |
> |----------|---|---------|-----------------|
> | DLL | 133 | GlibC, Linux, LiteOS, Zephyr, Course, SLING, SV-COMP | Doubly linked list |
> | SLL | 96 | Zephyr, Course, SLING, SV-COMP | Singly linked list |
> | Array | 8 | simple_array | Array |
> | Others | 23 | Course | Tree, hash table |
>
> **Extension rationale:** LIG-MM does not cover typical array operations, yet arrays are ubiquitous in real-world system software and require reasoning over contiguous memory addresses—fundamentally different from pointer-based structures. Neither LIG-SE nor SLING supports arrays. The added programs cover: element-wise operations, memory operations, structure conversions, and multi-loop operations.
>
> All programs involve at least one heap-allocated data structure and require shape invariants—none are purely numerical. We will add this to Section 4.1.
>
> ---
> ## Concern 2: Insufficient SFT Details
> > "A lot of details are missing and need to be added to the appendix. For instance, what data was the SFT done on?"
>
> **Training Data:**
> - **Source:** ICL refinement trajectories generated by MALICE (DeepSeek-V3.2 as teacher)
> - **Size:** 9,698 training samples distilled from successful trajectories
> - **Each sample:** Input = source code with previous (incorrect) invariant + QCP feedback (error type, location, trace at failure point, disproved entailment). Output = natural language **explanation** diagnosing root cause + concrete **correction** with revised invariant
> - **Generation:** For each program succeeding at iteration K, extract iterations 1–(K-1) as training examples
> - **Filtering:** Only trajectories leading to verified invariants are retained
>
> **Training Configuration:**
> - Base model: Meta-Llama-3.1-8B-Instruct
> - Fine-tuning: LoRA (rank 16, all linear layers)
> - Epochs: 3, lr: 2e-4 (cosine), warmup: 0.1
> - Validation: 5-fold cross-validation (programs grouped into folds to prevent leakage)
> - Hardware: 1× NVIDIA-5090, ~18h/fold
>
> **Result:** MALICE (SFT) achieves **86.15%** with only Refinement using 8B, surpassing full 685B ICL (82.69%), demonstrating that refinement knowledge can be effectively distilled into smaller models. We will expand in Section 4.5 and Appendix.
>
> ---
> ## Concern 3: Scalability with Code Size and Complexity
> > "There also could have been a discussion on how well the success of this method scales with increasing code size and complexity."
>
> Avg LoC is ~**36 lines/file** (260 programs, 9,197 total LoC), representative of modular functions in production codebases. We observe:
> - **Single-loop programs** with simple structures (array traversals, SLL iteration) achieve near-perfect pass rates
> - **Multi-loop programs** (e.g., `array_concat`) are more challenging but handled by MALICE's loop dependency tree (Algorithm 1)
> - **Failures** concentrate on complex pointer structures (double pointers), unsupported data structures (trees/hash tables without QCP strategies), and imprecise non-spatial conditions
>
> Scaling challenges: (1) **Shape generalization**—complex traces make pattern recognition harder; may require RL or additional context (predicate definitions, structural templates). (2) **Path explosion**—mitigated by K=4 unrolling and random sampling at depth 3–4.
>
> We will add a **Limitations** section discussing these considerations.
>
> ---
> ## Concern 4: Writing Quality
> > "The paper could use some copy editing. There are a number of typos and missing spaces."
>
> We will conduct a thorough copy-editing pass: fix quotation marks, missing spaces, incomplete sentence at line 57, bibliography capitalization (e.g., QCP), and other typographical errors.

---

> > ### Author Rebuttal · Reviewer_88vz · 2026-04-02
> >
> > Thank you, dear authors, for your comments.
> >
> > You have fully addressed my concerns 1, 2, and 4. For concern 3, it would be good to have a graph of code size (or some better complexity metric) and pass rate. I suggest adding such a graph.
> >
> > That being said, I will already increase my score to Weak Accept.

---

> > > ### Author Response · Authors · 2026-04-03
> > >
> > > We sincerely thank the reviewer for the thoughtful evaluation and for increasing the score to Weak Accept. We are grateful that our responses have fully resolved concerns 1, 2, and 4.
> > >
> > > Regarding **Concern 3**, we appreciate the constructive suggestion. We will add a figure in the revised paper plotting code complexity against pass rate. As a preview, we provide a breakdown by data structure category (which strongly correlates with structural complexity and average LoC):
> > >
> > > | Category | \# Programs | Avg LoC | MALICE (ICL) | MALICE (SFT) | SLING | LIG-SE |
> > > |---|---|---|---|---|---|---|
> > > | SLL | 96 | ~30 | **100.00%** | **100.00%** | 33.33% | 61.46% |
> > > | Array | 8 | ~21 | **100.00%** | 87.50% | 0.00% | 0.00% |
> > > | DLL | 133 | ~36 | 83.46% | **90.98%** | 24.81% | 32.33% |
> > > | Others (tree, hash) | 23 | ~35 | 0.00% | 0.00% | 13.04% | 30.43% |
> > > | **Overall** | **260** | **~35** | **82.69%** | **86.15%** | **26.15%** | **41.92%** |
> > >
> > > The trend is clear: MALICE achieves near-perfect pass rates on structurally simpler programs (SLL, Array), while DLL programs involving multi-pointer manipulation are more challenging (though SFT pushes this to 91%). The "Others" category (tree, hash table) fails across **all** methods due to missing predicate strategy implementations in the verifier QCP, not algorithmic limitations.
> > >
> > > In the camera-ready version, we will add a **scatter plot** of LoC vs. pass rate to provide a finer-grained scalability analysis as suggested.
> > >
> > > Thank you again for the valuable suggestion.

---

### Decision · Program_Chairs · 2026-04-30

**Decision:**

Accept (regular)

**Comment:**

This paper works on a long-standing challenge of generating shape loop invariants. The proposed approach MALICE includes two stages: a multi-step prediction stage that generates a predicate selection plan, and a second stage which is an agentic refinement stage, where the LLM iteratively debugs and corrects the failed loop invariants. Experimental results show noticeable improvement over existing traditional tools as well as several preliminary LLM-based approaches.

Overall the reviewers are satisfied with the empirical gain, where the framework can also have practical impact such as for C programs which is “a longstanding problem (reviewer 88vz)”.

I recommend acceptance of the paper, with the hope that the authors can further improve the paper clarity, and the clear statement of limitations and potential future directions as pointed out by reviewers.